# The Unbalanced Gromov Wasserstein Distance: Conic Formulation and Relaxation

**Thibault Séjourné**
Ecole Normale Supérieure, DMA, PSL
thibault.sejourne@ens.fr

**François-Xavier Vialard**
Université Gustave Eiffel
francois-xavier.vialard@u-pem.fr

**Gabriel Peyré**
Ecole Normale Supérieure, DMA, CNRS, PSL
gabriel.peyre@ens.fr

## Abstract

Comparing metric measure spaces (i.e. a metric space endowed with a probability distribution) is at the heart of many machine learning problems. The most popular distance between such metric measure spaces is the Gromov-Wasserstein (GW) distance, which is the solution of a quadratic assignment problem. The GW distance is however limited to the comparison of metric measure spaces endowed with a *probability* distribution. To alleviate this issue, we introduce two Unbalanced Gromov-Wasserstein formulations: a distance and a more tractable upper-bounding relaxation. They both allow the comparison of metric spaces equipped with arbitrary positive measures up to isometries. The first formulation is a positive and definite divergence based on a relaxation of the mass conservation constraint using a novel type of quadratically-homogeneous divergence. This divergence works hand in hand with the entropic regularization approach which is popular to solve large scale optimal transport problems. We show that the underlying non-convex optimization problem can be efficiently tackled using a highly parallelizable and GPU-friendly iterative scheme. The second formulation is a distance between mm-spaces up to isometries based on a conic lifting. Lastly, we provide numerical experiments on synthetic examples and domain adaptation data with a Positive-Unlabeled learning task to highlight the salient features of the unbalanced divergence and its potential applications in ML.

## 1 Introduction

Comparing data distributions on different metric spaces is a basic problem in machine learning. This class of problems is for instance at the heart of surfaces [Bronstein et al., 2006] or graph matching [Xu et al., 2019] (equipping the surface or graph with its associated geodesic distance), regression problems in quantum chemistry [Gilmer et al., 2017] (viewing the molecules as distributions of points in $\mathbb{R}^3$) and natural language processing [Grave et al., 2019, Alvarez-Melis and Jaakkola, 2018] (where texts in different languages are embedded as points distributions in different vector spaces).

**Metric measure spaces.** The mathematical way to formalize these problems is to model the data as *metric measure spaces* (mm-spaces). A mm-space is denoted as $\mathcal{X} = (X, d, \mu)$ where $X$ is a complete separable set endowed with a distance $d$ and a positive Borel measure $\mu \in \mathcal{M}_+(X)$. For instance, if $X = (x_i)_i$ is a finite set of points, then $\mu = \sum_i m_i \delta_{x_i}$ (here $\delta_{x_i}$ is the Dirac mass at $x_i$) is simply a set of positive weights $m_i = \mu(\{x_i\}) \geq 0$ associated to each point $x_i$, which accounts for its mass or importance. For instance, setting some $m_i$ to 0 is equivalent to removing the point $x_i$. We

35th Conference on Neural Information Processing Systems (NeurIPS 2021).

refer to Sturm [2012] for a mathematical account on the theory of mm-spaces. In all the applications highlighted above, it makes sense to perform the comparisons up to isometric transformations of the data. Two mm-spaces $\mathcal{X} = (X, d_X, \mu)$ and $\mathcal{Y} = (Y, d_Y, \nu)$ are considered to be equal (denoted $\mathcal{X} \sim \mathcal{Y}$) if they are isometric, meaning that there is a bijection $\psi : \mathrm{spt}(\mu) \to \mathrm{spt}(\nu)$ (where $\mathrm{spt}(\mu)$ is the support of $\mu$) such that $d_X(x, y) = d_Y(\psi(x), \psi(y))$ and $\psi_\sharp \mu = \nu$. Here $\psi_\sharp$ is the push-forward operator, so that $\psi_\sharp \mu = \nu$ is equivalent to imposing $\nu(A) = \mu(\psi^{-1}(A))$ for any set $A \subset Y$. For discrete spaces where $\mu = \sum_i m_i \delta_{x_i}$, then one should have $\nu = \psi_\sharp \mu = \sum_i m_i \delta_{\psi(x_i)}$. As highlighted by Mémoli [2011], considering mm-spaces up to isometry is a powerful way to formalize and analyze a wide variety of problems such as matching, regression and classification of distributions of points belonging to different spaces. Most often, the objects of interest come with a natural distance such as an intrinsic or extrinsic distance and the uniform measure is the usual choice to make mm-spaces widely applicable. The key to unlock all these problems is the computation of a distance between mm-spaces up to isometry. So far, existing distances (reviewed below) assume that $\mu$ is a probability distribution, i.e. $\mu(X) = 1$. This constraint is not natural and sometimes problematic for most of the practical applications to machine learning. The goal of this paper is to alleviate this restriction. We define for the first time a class of distances between unbalanced metric measure spaces, these distances being upper-bounded by divergences which can be approximated by an efficient numerical scheme.

**Csiszár divergences** The simplest case is when $X = Y$ and one simply ignores the underlying metric. One can then use Csiszár divergences (or $\varphi$-divergences), which perform a pointwise comparison (in contrast with optimal transport distances, which perform a displacement comparison). It is defined using an entropy function $\varphi : \mathbb{R}_+ \to [0, +\infty]$, which is a convex, lower semi-continuous, positive function with $\varphi(1) = 0$. The Csiszár $\varphi$-divergence reads $\mathrm{D}_\varphi(\mu|\nu) \triangleq \int_X \varphi\left(\frac{\mathrm{d}\mu}{\mathrm{d}\nu}\right)\mathrm{d}\nu + \varphi'_\infty \int_X \mathrm{d}\mu^\perp$, where $\mu = \frac{\mathrm{d}\mu}{\mathrm{d}\nu}\nu + \mu^\perp$ is called the Radon-Nikodym or the Lebesgue decomposition of $\mu$ with respect to $\nu$ and $\varphi'_\infty = \lim_{r\to\infty} \varphi(r)/r \in \mathbb{R} \cup \{+\infty\}$ is called the recession constant. This divergence $\mathrm{D}_\varphi$ is convex, positive, 1-homogeneous and weak* lower-semicontinuous, see Liero et al. [2015] for details. Particular instances of $\varphi$-divergences are Kullback-Leibler (KL) for $\varphi(r) = r \log(r) - r + 1$ (note that $\varphi'_\infty = \infty$) and Total Variation (TV) for $\varphi(r) = |r - 1|$.

**Balanced and unbalanced optimal transport.** If the common embedding space $X$ is equipped with a distance $d(x, y)$, one can use more elaborated methods such as optimal transport (OT) distances, which are computed by solving convex optimization problems. This type of methods has proven useful for ML problems as diverse as domain adaptation [Courty et al., 2014], supervised learning over histograms [Frogner et al., 2015] and unsupervised learning of generative models [Arjovsky et al., 2017]. In this case, the extension from probability distributions to arbitrary positive measures $(\mu, \nu) \in \mathcal{M}_+(X)^2$ is now well understood and corresponds to the theory of unbalanced OT. Following Liero et al. [2015], Chizat et al. [2018a], a family of unbalanced Wasserstein distances is defined by solving

$$\mathrm{UW}(\mu, \nu)^q \triangleq \inf_{\pi \in \mathcal{M}(X \times X)} \int \lambda(d(x, y))\mathrm{d}\pi(x, y) + \mathrm{D}_\varphi(\pi_1|\mu) + \mathrm{D}_\varphi(\pi_2|\mu). \quad (1)$$

Here $(\pi_1, \pi_2)$ are the two marginals of the joint distribution $\pi$, defined by $\pi_1(A) = \pi(A \times Y)$ for $A \subset X$. The mapping $\lambda : \mathbb{R}^+ \to \mathbb{R}$ and exponent $q \geq 1$ should be chosen wisely to ensure for instance that UW defines a distance (see Section 2.2.1). It is frequent to take $\rho\mathrm{D}_\varphi$ instead of $\mathrm{D}_\varphi$ (i.e. take $\psi = \rho\varphi$) to adjust the strength of the marginals' penalization. Balanced OT is retrieved with the convex indicator $\varphi = \iota_{\{1\}}$ (i.e. $\varphi(1) = 0$ and $\varphi(x) = +\infty$ otherwise) or by taking the limit $\rho \to +\infty$, which enforces $\pi_1 = \mu$ and $\pi_2 = \nu$. When $0 < \rho < +\infty$, unbalanced OT operates a trade-off between transportation and creation of mass, which is crucial to be robust to outliers in the data and to cope with mass variations in the modes of the distributions. For supervised tasks, the value of $\rho$ should be cross-validated to obtain the best performances. Its use is gaining popularity in applications, such as medical imaging registration [Feydy et al., 2019a], videos [Lee et al., 2019], generative learning [Balaji et al., 2020] and gradient flow to train neural networks [Chizat and Bach, 2018, Rotskoff et al., 2019]. Furthermore, existing efficient algorithms for balanced OT extend to this unbalanced problem. In particular Sinkhorn's iterations, introduced in ML for balanced OT by Cuturi [2013], extend to unbalanced OT [Chizat et al., 2018b, Séjourné et al., 2019], as detailed in Section 3.

**The Gromov-Wasserstein distance and its applications.** The Gromov-Wasserstein (GW) distance [Mémoli, 2011, Sturm, 2012] generalizes the notion of OT to the setting of mm-spaces up to

isometries. It replaces the linear cost $\int \lambda(d)\mathrm{d}\pi$ of OT by a quadratic function. It reads

$$\mathrm{GW}(\mathcal{X}, \mathcal{Y})^q \triangleq \min_{\pi \in \mathcal{M}_+(X \times Y)} \left\{ \int \lambda(|d_X(x,x') - d_Y(y,y')|)\mathrm{d}\pi(x,y)\mathrm{d}\pi(x',y') \; : \; \begin{matrix} \pi_1 = \mu \\ \pi_2 = \nu \end{matrix} \right\}. \quad (2)$$

It is proved in Mémoli [2011], Sturm [2012] that GW defines with $\lambda(t) = t^q$ a distance up to isometries on balanced mm-spaces (i.e. the measures are probability distributions). The GW distance is applied successfully in natural language processing for unsupervised translation learning [Grave et al., 2019, Alvarez-Melis and Jaakkola, 2018], in generative learning for objects lying in spaces of different dimensions [Bunne et al., 2019] and to build VAE for graphs [Xu et al., 2020]. It has been adapted for domain adaptation over different spaces [Redko et al., 2020]. It is also a relevant distance to compute barycenters between graphs or shapes [Vayer et al., 2018, Chowdhury and Needham, 2020]. When $(\mathcal{X}, \mathcal{Y})$ are Euclidean spaces, this distance compares distributions up to rigid isometry, and is closely related (but not equal) to metrics defined by procrustes analysis [Grave et al., 2019, Alvarez-Melis et al., 2019]. The problem (2) is non convex because the quadratic form $\int \lambda(|d_X - d_Y|)\mathrm{d}\pi \otimes \pi$ is not positive in general. It is in fact closely related to quadratic assignment problems [Burkard et al., 1998], which are used for graph matching problems, and are known to be NP-hard in general. Nevertheless, non-convex optimization methods have been shown to be successful in practice to use GW distances for ML problems. This includes for instance alternating minimization [Mémoli, 2011, Redko et al., 2020] and entropic regularization [Peyré et al., 2016, Gold and Rangarajan, 1996].

**Related works and contributions.** The concomitant work of De Ponti and Mondino [2020] extends the $L^p$ transportation distance defined in Sturm et al. [2006] to unbalanced mm-spaces and studies its geometric properties. This distortion distance is not equivalent to the GW distance, and is more difficult to estimate numerically because it explicitly imposes a triangle inequality constraint in the optimization problem. The work of Chapel et al. [2020] relaxes the GW distance to the unbalanced setting by hybridizing GW with partial OT [Figalli, 2010] for unsupervised labeling. It ressembles one particular setting of our formulation, but with some important differences, detailed in Section 2. Our construction is also connected to partial matching methods, which find numerous applications in graphics and vision [Cosmo et al., 2016]. In particular, Rodola et al. [2012] introduces a mass conservation relaxation of the GW problem.

The two main contributions of this paper are the definition of two formulations relaxing the GW distance. The first one is called the Unbalanced Gromov-Wasserstein (UGW) divergence and can be computed efficiently on GPUs. The second one is called the Conic Gromov-Wasserstein distance (CGW). It is proved to be a distance between mm-spaces endowed with positive measures up to isometries, as stated in Theorem 1 which is the main theoretical result of this paper. We also prove in Theorem 1 that UGW can be used as a surrogate upper-bounding CGW. We present those concepts and their properties in Section 2. We also detail in Section 3 an efficient computational scheme for a particular setting of UGW. This method computes an approximate stationary point of a biconvex relaxation of our formulations. Even though it is a lower bound of the original problem, we provide in Theorem 3 conditions ensuring the tightness of this relaxation in many cases of interest. The algorithm leverages the strength of entropic regularization and the Sinkhorn algorithm, namely that it is GPU-friendly and defines smooth loss functions amenable to back-propagation for ML applications. Section 4 provides some numerical experiments to highlight the qualitative behavior of this algorithm and its ability to cope with outliers and mass variations in the modes of the distributions. We illustrate numerically the tightness of the relation between UGW and CGW, showing that UGW is a reasonnable proxy of a distance, at least locally. We provide an application of our divergence in the positive unlabeled learning setting, using domain adaptation data, and display results which are at par or outperform the computable competitor Chapel et al. [2020].

## 2   Unbalanced Gromov-Wasserstein formulations

We present in this section our two new formulations and their properties. The first one, called UGW, is exploited in Sections 3 and 4 to derive an efficient algorithm used in numerical experiments. The second one, called CGW, defines a distance between mm-spaces up to isometries. Those results build upon the work of Liero et al. [2015], and a summary of the construction of UOT is detailed in Appendix A. In all what follows, we consider complete separable mm-spaces endowed with a metric and a positive measure.

## 2.1 The unbalanced Gromov-Wasserstein divergence

This new formulation makes use of quadratic $\varphi$-divergences, defined as $D_\varphi^\otimes(\rho|\nu) \triangleq D_\varphi(\rho \otimes \rho|\nu \otimes \nu)$, where $\rho \otimes \rho \in \mathcal{M}_+(X^2)$ is the tensor product measure defined by $d(\rho \otimes \rho)(x,y) = d\rho(x)d\rho(y)$. Note that $D_\varphi^\otimes$ is not a convex function in general.

**Definition 1** (Unbalanced GW). *The Unbalanced Gromov-Wasserstein divergence is defined as* $\text{UGW}(\mathcal{X}, \mathcal{Y}) = \inf_{\pi \in \mathcal{M}^+(X \times Y)} \mathcal{L}(\pi) \triangleq \mathcal{G}(\pi) + D_\varphi^\otimes(\pi_1|\mu) + D_\varphi^\otimes(\pi_2|\nu)$.

This definition can be understood as an hybridation between (1) and (2) but with a twist: one needs to use the quadratic divergence $D_\varphi^\otimes$ in place of $D_\varphi$. To the best of our knowledge, it is the first time such quadratic divergences are being used and studied. In the TV case, this is the most important distinction between UGW and partial-GW [Chapel et al., 2020]. Note also that the balanced GW distance (2) is recovered as a particular case when using $\varphi = \iota_{\{1\}}$ or by letting $\rho \to +\infty$ for an entropy $\psi = \rho\varphi$.

Using quadratic divergences results in UGW being 2-homogeneous: for $\theta \geq 0$, writing $(\mathcal{X}_\theta, \mathcal{Y}_\theta)$ equiped with $(\theta\mu, \theta\nu)$, one has $\theta^{-2}\text{UGW}(\mathcal{X}_\theta, \mathcal{Y}_\theta) = \text{UGW}(\mathcal{X}, \mathcal{Y})$. When using non tensorized $\varphi$-divergences, the resulting unbalanced Gromov-Wasserstein functional between $\mathcal{X}_\theta$ and $\mathcal{Y}_\theta$ have very different and inconsistent behaviors when $\theta \to 0$ and $\theta \to +\infty$. Indeed, once normalized by $\theta^{-2}$ and $\theta^{-1}$, one obtains respectively balanced GW and a Hellinger-type distance. Using tensorized divergences ensures that the behavior does not depends on $\theta$. It is also fundamental to connect UGW with our distance CGW, see Theorem 1 and Appendix B.

We first prove the existence of optimal plans $\pi$ minimizing $\mathcal{L}$, which holds for the three key settings of Section 2.2.1, namely for KL, TV, and for compact metric spaces (such as finite pointclouds and graphs). All proofs are deferred in Appendix B.

**Proposition 1** (Existence of minimizers). *We assume that $(X, Y)$ are compact and that either (i) $\varphi$ superlinear, i.e $\varphi'_\infty = \infty$, or (ii) $\lambda$ has compact sublevel sets in $\mathbb{R}_+$ and $2\varphi'_\infty + \inf \lambda > 0$. Then there exists $\pi \in \mathcal{M}_+(X \times Y)$ such that $\text{UGW}(\mathcal{X}, \mathcal{Y}) = \mathcal{L}(\pi)$.*

The following proposition ensures that the functional UGW can be used to compare mm-spaces.

**Proposition 2** (Definiteness of UGW). *Assume that $\varphi^{-1}(\{0\}) = \{1\}$ and $\lambda^{-1}(\{0\}) = \{0\}$. Then $\text{UGW}(\mathcal{X}, \mathcal{Y}) \geq 0$ and is $0$ if and only if $\mathcal{X} \sim \mathcal{Y}$.*

We end this section with a reformulation of UGW which is important to make the connection with the second formulation CGW of the following section. It splits UGW into two parts: the term $\varphi(0)(|(\mu \otimes \mu)^\perp| + |(\nu \otimes \nu)^\perp|)$ accounts for the pure creation/destruction of mass and a new transport cost $L_c$ accounts for the remaining part (partial/pure transport and partial creation/destruction of mass).

**Lemma 1.** *Defining $L_c(a,b) \triangleq c + a\varphi(1/a) + b\varphi(1/b)$, and writing $(f \triangleq \frac{d\mu}{d\pi_1}, g \triangleq \frac{d\nu}{d\pi_2})$ the Lebesgue densities of $(\mu, \nu)$ w.r.t. $(\pi_1, \pi_2)$ such that $\mu = f\pi_1 + \mu^\perp$ and $\nu = g\pi_2 + \nu^\perp$, one has*

$$\mathcal{L}(\pi) = \int_{X^2 \times Y^2} L_{\lambda(|d_X - d_Y|)}(f \otimes f, g \otimes g)d\pi d\pi + \varphi(0)(|(\mu \otimes \mu)^\perp| + |(\nu \otimes \nu)^\perp|). \quad (3)$$

*Proof.* Write $f = \frac{d\mu}{d\pi_1}$ and $g = \frac{d\nu}{d\pi_2}$. The Lebesgue decompositions read $\mu \otimes \mu = (f \otimes f)\pi_1 \otimes \pi_1 + (\mu \otimes \mu)^\perp$ and $\nu \otimes \nu = (g \otimes g)\pi_2 \otimes \pi_2 + (\nu \otimes \nu)^\perp$, thanks to the tensorized structure of the decomposed plans. To prove Equation (3), we need to define the reverse entropy Liero et al. [2015] such that $D_\varphi(\alpha|\mu) = D_\psi(\mu|\alpha)$, where $\psi(x) \triangleq x\varphi(\frac{1}{x})$ is also an entropy function satisfying $\psi'_\infty = \varphi(0)$. One then has

$$\mathcal{L}(\pi) = \int_{X^2 \times Y^2} \lambda(\Gamma)d\pi d\pi + D_\varphi^\otimes(\pi_1|\mu) + D_\varphi^\otimes(\pi_2|\nu)$$

$$= \int_{X^2 \times Y^2} \lambda(\Gamma)d\pi d\pi + D_\psi^\otimes(\mu|\pi_1) + D_\psi^\otimes(\nu|\pi_2)$$

$$\mathcal{L}(\pi) = \int_{X^2 \times Y^2} \lambda(\Gamma) \mathrm{d}\pi \mathrm{d}\pi + \int_{X^2} \psi(f \otimes f) \mathrm{d}\pi_1 \mathrm{d}\pi_1 + \int_{Y^2} \psi(g \otimes g) \mathrm{d}\pi_2 \mathrm{d}\pi_2$$
$$+ \varphi(0)(|(\mu \otimes \mu)^\perp| + |(\nu \otimes \nu)^\perp|)$$
$$= \int_{X^2 \times Y^2} L_{\lambda(\Gamma)}(f \otimes f, g \otimes g) \mathrm{d}\pi \mathrm{d}\pi + \varphi(0)(|(\mu \otimes \mu)^\perp| + |(\nu \otimes \nu)^\perp|).$$

Using the definition of $\psi$ in $L_c$ ends the proof. $\qquad \square$

## 2.2 The conic Gromov-Wasserstein distance

We introduce a second "conic" formulation of unbalanced GW, which is connected to UGW, and whose construction is inspired by the conic formulation of UOT (see Appendix A for an overview).

### 2.2.1 Background on cone sets and distances

The conic formulation lifts a point $x \in X$ to a couple $(x, r) \in X \times \mathbb{R}^+$ where $r$ encodes some (power of a) mass. Then we seek optimal transport plans defined over $\mathfrak{C}[X] \triangleq X \times \mathbb{R}_+/(X \times \{0\})$, where coordinates $(x, r = 0)$ with no mass are merged into a single point $\mathfrak{o}_X$ called the apex of the cone. In the sequel, points of $X \times \mathbb{R}_+$ are noted $(x, r)$, while $[x, r]$ are quotiented points of $\mathfrak{C}[X]$.

While transport plans depend on variables $([x, r], [y, s])$ and $([x', r'], [y', s'])$ in $\mathfrak{C}[X] \times \mathfrak{C}[Y]$, the transportation cost involved in our conic formulation only makes use of the 2-D cone $\mathfrak{C}[\mathbb{R}_+]$ over $\mathbb{R}_+$ endowed with the distance $|u - v|$ (note that any other distance on $\mathbb{R}$ could be used as well). More specifically, we consider coordinates of the form $([u, a], [v, b]) = ([d_X(x, x'), rr'], [d_Y(y, y'), ss']) \in \mathfrak{C}[\mathbb{R}_+] \times \mathfrak{C}[\mathbb{R}_+]$. Thus we now describe conic discrepancies $\mathcal{D}$ on $\mathfrak{C}[\mathbb{R}_+]$, which are defined for $(p, q) \geq 1$ as $\mathcal{D}([u, a], [v, b])^q \triangleq H_{\lambda(|u-v|)}(a^p, b^p)$, where $H_c(a^p, b^p) \triangleq \inf_{\theta \geq 0} \theta L_c(\frac{a^p}{\theta}, \frac{b^p}{\theta})$ is the perspective transform of $L_c$ introduced in Lemma 1. The intuition underpinning the definition of this cost is that the perspective transform accounts for the possibility to rescale a transport plan $\pi$ by a scalar $\theta$ but the scaling is performed pointwise instead of globally. In general $\mathcal{D}$ is not a distance, but it is always definite as stated by this result proved in Appendix A.

**Proposition 3.** *Assume $\lambda^{-1}(\{0\}) = \{0\}$, $\varphi^{-1}(\{0\}) = \{1\}$ and $\varphi$ is coercive. Then $\mathcal{D}$ is definite on $\mathfrak{C}[\mathbb{R}^+]$, i.e. $\mathcal{D}([u, a], [v, b]) = 0$ if and only if $(a = b = 0)$ or $(a = b$ and $u = v)$.*

Of particular interest are those $\varphi$ where $\mathcal{D}$ is a distance, which necessitates a careful choice of $\lambda, p$ and $q$. We now detail three examples where this is the case.

**Gaussian Hellinger distance (GH).** When $\mathrm{D}_\varphi = \mathrm{KL}$, $\lambda(t) = t^2$ and $q = p = 2$, then one has $\mathcal{D}([u, a], [v, b])^2 = a^2 + b^2 - 2abe^{-|u-v|/2}$. This cone distance [Burago et al., 2001] is further generalized by De Ponti [2019] who shows that $\mathcal{D}$ is a distance for power entropies $\varphi(s) = \frac{s^p - p(s-1) - 1}{p(p-1)}$ if $p \geq 1$ (the case $p = 1$ corresponding to $\mathrm{D}_\varphi = \mathrm{KL}$).

**Hellinger-Kantorovich (HK) / Wasserstein-Fisher-Rao distance (WFR).** When $\mathrm{D}_\varphi = \mathrm{KL}$, $\lambda(t) = -\log \cos^2(t \wedge \frac{\pi}{2})$ and $q = p = 2$, then one has $\mathcal{D}([u, a], [v, b])^2 = a^2 + b^2 - 2ab \cos(\frac{\pi}{2} \wedge |u - v|)$. This construction, which might seem peculiar, corresponds to the one used to make unbalanced OT a geodesic distance, as detailed in [Liero et al., 2015, Chizat et al., 2018a].

**Partial optimal transport distance (PT).** When $\mathrm{D}_\varphi = \mathrm{TV}$, $\lambda(t) = t^q$, $q \geq 1$ and $p = 1$, then $\mathcal{D}([u, a], [v, b])^q = a + b - (a \wedge b)(2 - |u - v|^q)_+$ defines a cone distance [Chizat et al., 2018a].

### 2.2.2 Definitions and properties

The conic formulation consists in solving a GW problem on the cone, with the addition of two linear constraints. Informally speaking, $L_c$ from Lemma 1 becomes $\mathcal{D}$, the term $(|(\mu \otimes \mu)^\perp| + |(\nu \otimes \nu)^\perp|)$ is taken into account by the constraints (5) below, and the variables $(f, g)$ are replaced by $(r^p, s^p)$. It reads $\mathrm{CGW}(\mathcal{X}, \mathcal{Y}) \triangleq \inf_{\alpha \in \mathcal{U}_p(\mu, \nu)} \mathcal{H}(\alpha)$ where

$$\mathcal{H}(\alpha) \triangleq \int \mathcal{D}([d_X(x, x'), rr'], [d_Y(y, y'), ss'])^q \, \mathrm{d}\alpha([x, r], [y, s]) \mathrm{d}\alpha([x', r'], [y', s']), \quad (4)$$

and $\mathcal{U}_p(\mu, \nu)$ is defined as the set

$$\mathcal{U}_p(\mu, \nu) \triangleq \left\{ \alpha \in \mathcal{M}_+(\mathfrak{C}[X] \times \mathfrak{C}[Y]), \int_{\mathbb{R}_+} r^p \mathrm{d}\alpha_1(\cdot, r) = \mu, \int_{\mathbb{R}_+} s^p \mathrm{d}\alpha_2(\cdot, s) = \nu \right\}. \quad (5)$$

It is similar to the conic formulation of UW, see Appendix A. Note that similarly to the GW formulation (2) – and in sharp contrast with the conic formulation of UW – here the transport plans are defined on the cone $\mathfrak{C}[X] \times \mathfrak{C}[Y]$ but the cost $\mathcal{D}$ is a distance on $\mathfrak{C}[\mathbb{R}_+]$.

We present now the main contributions of this paper, proved in Appendix C. We state that CGW defines a distance under conditions that hold for the settings of Section 2.2.1, and that it is upper-bounded by UGW. The divergence UGW can be approximated with efficient numerical schemes as detailed in Section 3.

**Theorem 1.** *(i) The divergence* CGW *is symmetric, positive and definite up to isometries. (ii) If $\mathcal{D}$ is a distance on $\mathfrak{C}[\mathbb{R}_+]$, then* CGW$^{1/q}$ *is a distance on the set of mm-spaces up to isometries. (iii) For any $(\mathrm{D}_\varphi, \lambda, p, q)$ with associated cost $\mathcal{D}$ on the cone, one has* UGW $\geq$ CGW.

## 3 Algorithms

We focus in this section on the numerical computation of the upper bound UGW using a bi-convex relaxation and derive an alternate minimization scheme coupled with entropic regularization. We also propose to approximate CGW by doing a similar alternate minimization, as detailed in Appendix E. We provide guarantees of tightness on the bi-convex relaxation for CGW (see Theorem 3). The computation of the distance CGW is heavy in practice because it requires an optimization over a lifted conic space, which needs to be discretized. Thus it does not scale to large problem for CGW, but allows to explore numerically how tight is the upper bound UGW $\geq$ CGW, see Section 4. The algorithm for UGW is presented on arbitrary measures, the special case of discrete measures being a particular case. The discretized formulas and algorithms are detailed in Appendix D, see also Chizat et al. [2018b], Peyré et al. [2016]. All implementations are available at https://github.com/thibsej/unbalanced_gromov_wasserstein, and installable in Python with the command *pip install unbalancedgw*.

### 3.1 Bi-convex relaxation and tightness

In order to derive a simple numerical approximation scheme, following Mémoli [2011], we introduce a lower bound obtained by introducing two transportation plans. To further accelerate the method and enable GPU-friendly iterations, similarly to Gold et al. [1996], Solomon et al. [2016], we consider an entropic regularization. It reads, for any $\varepsilon \geq 0$,

$$\mathrm{UGW}_\varepsilon(\mathcal{X}, \mathcal{Y}) \triangleq \inf_\pi \mathcal{L}(\pi) + \varepsilon \mathrm{KL}^\otimes(\pi | \mu \otimes \nu) \geq \inf_{\pi, \gamma} \mathcal{F}(\pi, \gamma) + \varepsilon \mathrm{KL}(\pi \otimes \gamma | (\mu \otimes \nu)^{\otimes 2}), \tag{6}$$

$$\text{and} \quad \mathcal{F}(\pi, \gamma) \triangleq \int_{X^2 \times Y^2} \lambda(|d_X - d_Y|) \mathrm{d}\pi \otimes \gamma + \mathrm{D}_\varphi(\pi_1 \otimes \gamma_1 | \mu \otimes \mu) + \mathrm{D}_\varphi(\pi_2 \otimes \gamma_2 | \nu \otimes \nu),$$

where $(\gamma_1, \gamma_2)$ denote the marginals of the plan $\gamma$. In the sequel we write $\mathcal{F}_\varepsilon = \mathcal{F} + \varepsilon \mathrm{KL}^\otimes$. Note that in contrast to the entropic regularization of GW Peyré et al. [2016], here we use a tensorized entropy to maintain the overall homogeneity of the energy. A simple method to approximate this lower bound is to perform an alternate minimization on $\pi$ and $\gamma$, which is known to converge for smooth $\varphi$ to a stationary point since the coupling term in the functional is smooth [Tseng, 2001]. Note that if $\pi \otimes \gamma$ is optimal then so is $(s\pi) \otimes (\frac{1}{s}\gamma)$ with $s \geq 0$. Thus without loss of generality we can optimize under the constraint $m(\pi) = m(\gamma)$ by setting $s = \sqrt{m(\gamma)/m(\pi)}$.

We now discuss the tightness of the bi-convex relaxation by generalizing a result of Konno. We first present a result which applies to general quadratic assignment problems, then state its application to our setting.

**Theorem 2** (Tight relaxation). *Let $B$ a Banach space, let $f : B \mapsto \mathbb{R} \cup \{+\infty\}$ be a function and let $\mathcal{L} : C \subset B \mapsto \mathbb{R}$ the function defined on the convex set $C \subset B$ by $\mathcal{L}(\pi) = \frac{1}{2}\langle \pi, k(\pi) \rangle + 2f(\pi)$ where $k$ is a symmetric bilinear map which is negative (not necessarily definite) on $\Delta C \triangleq \mathrm{Span}(\{\pi - \gamma ; (\pi, \gamma) \in C\})$, that is, for any $z \in \Delta C$, $\langle z, kz \rangle \leq 0$. Assume that there exists $\pi_0 \in C$ such that $\mathcal{L}(\pi_0) < +\infty$, and define $\mathcal{F}(\pi, \gamma) \triangleq \frac{1}{2}\langle \pi, k(\gamma) \rangle + f(\pi) + f(\gamma)$. Then, for any $(\pi_*, \gamma_*) \in \arg\min \mathcal{F}(\pi, \gamma)$, we have $\mathcal{F}(\pi_*, \pi_*) = \mathcal{F}(\gamma_*, \gamma_*) = \mathcal{F}(\pi_*, \gamma_*)$. Moreover, if one assumes either that $k$ is a definite kernel or $f$ is strictly convex, one gets $\pi_* = \gamma_*$.*

The above Theorem 2 is proved in Appendix D. As an application, we now state our tightness result for $\mathrm{GW}_\varepsilon$ and CGW. In those settings the optimizers of the bi-convex relaxation are also optimal for the original problem.

**Theorem 3.** *For $GW_\varepsilon$ with $\varepsilon \geq 0$ or for CGW, assume that $\lambda(t) = t^2$ and that $(d_X, d_Y)$ are both conditionnally negative (or conditionally positive) kernels. Then the bi-convex relaxation of both problems is tight.*

*Proof.* The proof for CGW is detailed in Appendix E, we prove the tightness for $\mathrm{GW}_\varepsilon$. When $\lambda(t) = t^2$ the kernel $k = \lambda(|d_X - d_Y|)$ is conditionally negative on the set $\{(\pi, \gamma), \ \pi_1 = \gamma_1 \text{ and } \pi_2 = \gamma_2\}$, i.e. we have $\langle (\pi - \gamma), \ k(\pi - \gamma) \rangle \leq 0$ (see Maron and Lipman [2018]). For $\mathrm{GW}_\varepsilon$ one has $\pi_1 = \gamma_1 = \mu$ and $\pi_2 = \gamma_2 = \nu$ thanks to the constraints on marginals. Thus the kernel is negative semi-definite and the proof of Theorem 2 applies, hence the tightness of the relaxation. $\square$

Konno's result Konno [1976] applies for unregularized ($\varepsilon = 0$), Balanced-GW. The novelty of Theorem 3 is its extension to both $\mathrm{GW}_\varepsilon$ and CGW. So far it is an open question whether the relaxation is tight or not for $\mathrm{UGW}_\varepsilon$, because the above proof no longer holds. Note that in all our numerical simulations, our solvers always found solutions of $\mathrm{UGW}_\varepsilon$ such that $\pi = \gamma$ when $|d_X - d_Y|^2$ is conditonally negative. The property that the kernel $|d_X - d_Y|^2$ is negative does not hold in general (e.g. for graph geodesic distances) and the tightness of the relaxation remains open in this setting. We know from [Maron and Lipman, 2018, Theorem 1] that it is conditionally negative semi-definite when both $(d_X, d_Y)$ are conditionally negative kernels. Examples of distances which are negative kernels are tree metrics in the case of graphs, as well as Euclidean, spherical and hyperbolic distances over their respective manifolds Feragen et al. [2015]. In practice, when $\varepsilon$ is small, we observed in the indefinite setting that the relaxation outputs more frequently spurious minima than in the negative semi-definite setting.

## 3.2 Alternate Sinkhorn minimization

Minimizing the lower bound (6) with respect to either $\pi$ or $\gamma$ is non-trivial for an arbitrary $\varphi$. We restrict our attention to the Kullback-Leibler case $\mathrm{D}_\varphi = \rho \mathrm{KL}$ with $\rho > 0$, which can be addressed by solving a regularized and convex unbalanced problem as studied in Chizat et al. [2018b], Séjourné et al. [2019]. It is explained in the following proposition.

**Proposition 4.** *For a fixed $\gamma$, the optimal $\pi \in \arg\min_\pi \mathcal{F}(\pi, \gamma) + \varepsilon\mathrm{KL}(\pi \otimes \gamma|(\mu \otimes \nu)^{\otimes 2})$ solves*

$$\min_\pi \int c_\gamma^\varepsilon(x, y)\mathrm{d}\pi(x, y) + \rho m(\gamma)\mathrm{KL}(\pi_1|\mu) + \rho m(\gamma)\mathrm{KL}(\pi_2|\nu) + \varepsilon m(\gamma)\mathrm{KL}(\pi|\mu \otimes \nu),$$

*where $m(\gamma) \triangleq \gamma(X \times Y)$ is the mass of $\gamma$, and where we define the cost associated to $\gamma$ as*

$$c_\gamma^\varepsilon(x, y) \triangleq \int \lambda(|d_X(x, \cdot) - d_Y(y, \cdot)|)\mathrm{d}\gamma + \rho\int \log(\frac{\mathrm{d}\gamma_1}{\mathrm{d}\mu})\mathrm{d}\gamma_1 + \rho\int \log(\frac{\mathrm{d}\gamma_2}{\mathrm{d}\nu})\mathrm{d}\gamma_2 + \varepsilon\int \log(\frac{\mathrm{d}\gamma}{\mathrm{d}\mu\mathrm{d}\nu})\mathrm{d}\gamma.$$

Computing the cost $c_\gamma^\varepsilon$ for spaces $X$ and $Y$ of $n$ points has in general a cost $O(n^4)$ in time and memory. However, as explained for instance in Peyré et al. [2016], for the special case $\lambda(t) = t^2$, this cost is reduced to $O(n^3)$ in time and $O(n^2)$ in memory. This is the setting we consider in the numerical simulations. This makes the method applicable for scales of the order of $10^4$ points. For larger datasets one should use approximation schemes such as hierarchical approaches [Xu et al., 2019] or Nyström compression of the kernel [Altschuler et al., 2018].

The resulting alternate minimization method is detailed in Algorithm 1, see Appendix D for a discretized version. It uses the unbalanced Sinkhorn algorithm of Chizat et al. [2018b], Séjourné et al. [2019] as sub-iterations and takes $\pi = \mu \otimes \nu/\sqrt{m(\mu)m(\nu)}$ to initialize the updates. This Sinkhorn algorithm operates over a pair of continuous functions (so-called

---

**Algorithm 1 – UGW$(\mathcal{X}, \mathcal{Y}, \rho, \varepsilon)$**

**Input:** mm-spaces $(\mathcal{X}, \mathcal{Y})$, relax. $\rho$, regul. $\varepsilon$
**Output:** $\pi, \gamma$ solving (6)

Init. $\pi = \gamma = \mu \otimes \nu/\sqrt{m(\mu)m(\nu)}$, $g = 0$.
**while** $(\pi, \gamma)$ has not converged **do**
   Update $\pi \leftarrow \gamma$,
   then $c \leftarrow c_\pi^\varepsilon$, $\tilde{\rho} \leftarrow m(\pi)\rho$, $\tilde{\varepsilon} \leftarrow m(\pi)\varepsilon$
   **while** $(f, g)$ has not converged **do**
     $f \leftarrow -\frac{\tilde{\varepsilon}\tilde{\rho}}{\tilde{\varepsilon}+\tilde{\rho}} \log \int e^{(g(y)-c(\cdot, y))/\tilde{\varepsilon}}\mathrm{d}\nu(y)$
     $g \leftarrow -\frac{\tilde{\varepsilon}\tilde{\rho}}{\tilde{\varepsilon}+\tilde{\rho}} \log \int e^{(f(x)-c(x, \cdot))/\tilde{\varepsilon}}\mathrm{d}\mu(x)$
   **end while**
   Upd. $\gamma(x, y) \leftarrow e^{\frac{f(x)+g(y)-c(x,y)}{\tilde{\varepsilon}}}\mu(x)\nu(y)$
   Rescale $\gamma \leftarrow \sqrt{m(\pi)/m(\gamma)}\gamma$
**end while**
Return $(\pi, \gamma)$.

---

Kantorovitch potentials) $f(x)$ and $g(y)$. For discrete spaces $X$ and $Y$ of size $n$, these functions are stored in vectors of size $n$, and that integral involved in the updates becomes a sum. Each iteration of Sinkhorn thus has a cost $n^2$, and all the involved operation can be efficiently mapped to parallelizable GPU routines as detailed in Chizat et al. [2018b], Séjourné et al. [2019]. Another advantage of using an unbalanced Sinkhorn algorithm is its complexity $O(n^2/\varepsilon)$ to compute an $\varepsilon$-approximation, as stated in Pham et al. [2020], which should be compared to $O(n^2/\varepsilon^2)$ operations for balanced Sinkhorn.

Note also that balanced GW is recovered as a special case when setting $\rho \to +\infty$, so that $\tilde{\rho}/(\tilde{\varepsilon} + \tilde{\rho}) \to 1$ should be used in the iterations. In order to speed up Sinkhorn inner-loops, especially for small values of $\varepsilon$, one can use linear extrapolation [Thibault et al., 2017] or non-linear Anderson acceleration [Anderson, 1965, Scieur et al., 2016].

There is an extra scaling step after computing $\gamma$ involving the mass $m(\pi)$. It corresponds to the scaling $s$ of $\pi \otimes \gamma$ such that $m(\pi) = m(\gamma)$, and we observe that this scaling is key not only to impose this mass equality but also to stabilize the algorithm. Otherwise we observed that $m(\gamma) < 1 < m(\pi)$ and underflows whenever $m(\gamma) \to 0$ and $m(\pi) \to \infty$.

## 4    Numerical experiments

This section presents simulations on synthetic examples to highlight the qualitative behavior of UGW and the tightness of the bound UGW $\geq$ CGW. Other illustrations on UGW are available in Appendix E. We end the section with a learning application of UGW in a positive-unlabeled setting, using domain adaptation data so as to compare with PGW Chapel et al. [2020]. In the synthetic experiments, $\mu$ and $\nu$ are probability distributions, which allows us to compare GW with UGW.

**Robustness to imbalanced classes.**    In this first example, we take $X = \mathbb{R}^3$, $Y = \mathbb{R}^2$ and consider $\mathcal{E}_2, \mathcal{E}_3, \mathcal{C}$ and $\mathcal{S}$ to be uniform distributions on a 2D and 3D ellipse, a square and a sphere. We consider mm-spaces of different dimensions to emphasize the ability of (U)GW to compare different spaces. Figure 1 contrasts the transportation plan obtained by GW and UGW for a fixed $\mu = 0.5\mathcal{E}_3 + 0.5\mathcal{S}$ and $\nu$ obtained using two different mixtures of $\mathcal{E}_2$ and $\mathcal{C}$. The black segments show the largest entries of the transportation matrix $\pi$, for a sub-sampled set of points (to ease visibility), thus effectively displaying the matching induced by the plan. Furthermore, the width of the dots are scaled according to the mass of the marginals $\pi_1 \approx \mu$ and $\pi_2 \approx \nu$, i.e. the smaller the point, the smaller is the amount of transported mass. This figure shows that the exact conservation of mass imposed by GW leads to a poor geometrical matching of the shapes which have different global mass. As this should be expected, UGW recovers coherent matchings. We suspect the alternate minimization algorithm is able to find the global minimum in these cases.

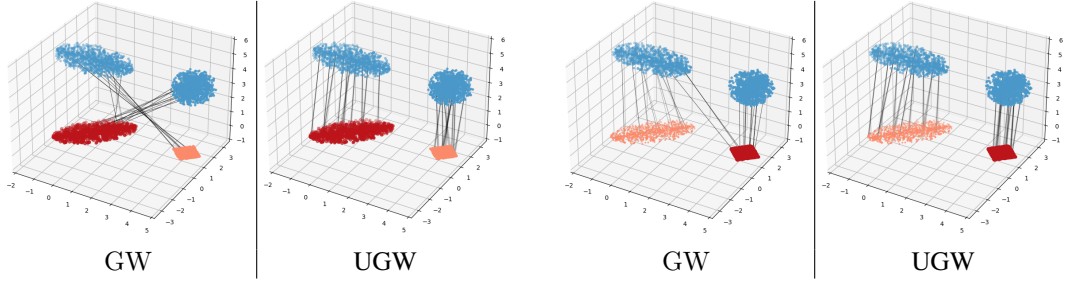

GW UGW GW UGW

Figure 1: GW vs. UGW transportation plan, using $\nu = 0.3\mathcal{E}_2 + 0.7\mathcal{C}$ on the left, and $\nu = 0.7\mathcal{E}_2 + 0.3\mathcal{C}$ on the right. The 2D mm-spaces is lifted into $\mathbb{R}^3$ by padding the third coordinate to zero.

**Tightness of the bound CGW$\leq$UGW**    We propose to approximate CGW by doing a similar alternate minimization as for UGW, as detailed in Appendix E. This numerical scheme does not scale to large problems, but allows us to explore numerically how tight is the upper bound UGW $\geq$ CGW. Figure 2 highlights the fact that in Euclidean space $X = Y = \mathbb{R}^d$, this bound seems to be tight when the two measures are sufficiently close. We consider discrete measures $\mu = \frac{1}{n}\sum_i \delta_{x_i}$ in $X = Y = \mathbb{R}^d$ and $\nu_t = \frac{1}{n}\sum_i \delta_{y_i}$ where $y_i = x_i + t\Delta_i$ where $\Delta_i$ are random perturbations and

denote $(\mathcal{X}, \mathcal{Y}_t)$ the two mm-spaces associated to the Euclidean distance. As $t \to 0$, $\mu$ and $\nu_t$ get closer, we observe numerically that UGW $\approx$ CGW. Figure 3 considers random points $(x_i)_i$ and $(y_i)_i$ and displays the histograms of the ratio CGW/UGW for $n = 3$. This shows that while the bound CGW $\leq$ UGW seems not tight, the ratio appears to be bounded even for points not being close. This numerical experiment suggests that UGW and CGW are locally equivalent and that UGW is in practice an acceptable proxy of the distance CGW. We leave for future works a tighter analysis of the gap between UGW and CGW.

**Positive unlabeled learning experiments**  Positive Unlabeled (PU) learning is a semi-supervised classification problem, where instead of learning from positive and negative samples $(x_i, \ell_i)_i$ with labels $\ell_i \in \{-1, 1\}$ we only learn from one class labeled with positives, i.e. only those $X \triangleq \{x_i : \ell_i = 1\}$. The task is to leverage $X$ to predict the classes $\ell = \ell(y) \in \{-1, +1\}$ of unlabelled $y \in Y$ belong to a separate space. We consider here that $X, Y$ are embedded in Euclidean space, and denote $\mathcal{X}, \mathcal{Y}$ the associated labelled and unlabelled mm-spaces, equipped with the uniform distribution. Our experiments are adapted from Partial-GW (PGW) Chapel et al. [2020], which used partial GW to solve PU-learning. The rationale of using unbalanced OT methods for PU learning stems from the fact that positive samples should be matched with positive due to their similar features, while negative samples would be ignored due to dissimilar features that induce a laziness to transport mass and match them.

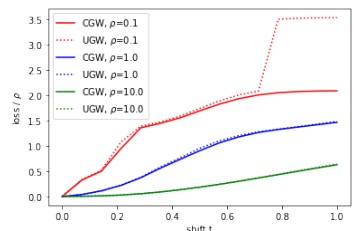

Figure 2: Comparison of $\text{UGW}(\mathcal{X}, \mathcal{Y}_t)$ and $\text{CGW}(\mathcal{X}, \mathcal{Y}_t)$ as the support gets shifted by a perturbation.

We consider PU learning over the Caltech office dataset used for domain adaptation tasks (with domains Caltech (C) Griffin et al. [2007], Amazon (A), Webcam (W) and DSLR (D) Saenko et al. [2010]). The Caltech datasets are represented with two embeddings based on Surf and Decaf features Saenko et al. [2010], Donahue et al. [2014]. On the latter datasets, we perform PU learning over similar features (e.g. surf-C → surf-* or decaf-C → decaf-*) and from one feature format to the other (e.g. surf-C → decaf-* or surf-C → decaf-*). Those features are projected via PCA to subspaces of dimension 10 for surf features and 40 for decaf features. In the last task, one cannot use standard PU-method, and to the best of our knowledge, Unbalanced-GW methods are the only approaches for PU learning across different domains/features.

The procedure is the following. We solve the PU learning problem by computing the optimal plan $\pi$ for $\text{UGW}(\mathcal{X}, \mathcal{Y})$. We compute its first marginal $\pi_2$ on $\mathcal{Y}$, and predict the labels of some $y \in Y$ as $\ell(y) \triangleq \text{sign}(\pi_2(y) - q)$ where $q$ is the quantile of $\pi_2$ corresponding to the proportion $r$ of positives samples in $Y$. Following Chapel et al. [2020] which is adapted from Kato et al. [2018], Hsieh et al. [2019], this proportion $r$ is assumed to be known. We report the accuracy of the prediction over the same 20 folds of the datasets, and use 20 other folds to validate the parameters of UGW. We consider 100 random samples for each fold of $(X, Y)$, a ratio of positive samples $r = 0.1$ for domains (C,A,W,D), and a ratio $r = 0.2$ for domains (C,A,W).

Figure 3: Histograms of the ratio CGW/UGW for random spaces with $n \in \{2, 3, 5\}$ samples. Ratios over 1 are due to local minima.

Since the GW objective is non-convex, the initialization of the minimization algorithms is key to obtain good performances. In Chapel et al. [2020] and our experiments, for tasks where $Y$ and $Y$ belong to the same Euclidean space, (e.g. surf-C → surf-*) we initialize $\pi$ with the Partial-Wasserstein (PW) solution with a squared Euclidean cost. For cross-domain prediction (e.g. surf-C → decaf-*), following Chapel et al. [2020], PGW is initialized with a list of plans built using a coarsened representation of the data with $k$-NN. While Chapel et al. [2020] makes use in an oracle manner of the plan providing the best accuracy, we modified their protocol and keep the plan which has the lowest PGW cost, which seems fairer, hence the difference in performance with Chapel et al. [2020]. To initialize UGW when $X$ and $Y$ do not belong to the same Euclidean space, we use a UOT solution of a matching between distance

| Dataset | prior | Init (PW) | PGW | **UGW** | Dataset | prior | Init (FLB) | PGW | **UGW** |
|---|---|---|---|---|---|---|---|---|---|
| surf-C → surf-C | 0.1 | **89.9** | 84.9 | 83.9 | surf-C → decaf-C | 0.1 | 85.0 | 85.1 | **85.6** |
| surf-C → surf-A | 0.1 | 81.8 | 82.2 | **83.5** | surf-C → decaf-A | 0.1 | 84.2 | **87.1** | 83.6 |
| surf-C → surf-W | 0.1 | **81.9** | 81.3 | 80.3 | surf-C → decaf-W | 0.1 | 86.2 | **88.6** | 86.8 |
| surf-C → surf-D | 0.1 | 80.0 | 81.4 | **83.2** | surf-C → decaf-D | 0.1 | 84.7 | **91.1** | 90.7 |
| surf-C → surf-C | 0.2 | **79.7** | 75.7 | 75.4 | surf-C → decaf-C | 0.2 | 74.8 | 75.6 | **75.9** |
| surf-C → surf-A | 0.2 | 65.6 | 66.0 | **76.4** | surf-C → decaf-A | 0.2 | 76.2 | **87.9** | 82.4 |
| surf-C → surf-W | 0.2 | 65.1 | 64.3 | **67.3** | surf-C → decaf-W | 0.2 | 81.5 | 88.4 | **89.9** |
| decaf-C → decaf-C | 0.1 | **93.9** | 83.0 | 86.8 | decaf-C → surf-C | 0.1 | **81.7** | 81.0 | 81.1 |
| decaf-C → decaf-A | 0.1 | 80.1 | 81.4 | **85.6** | decaf-C → surf-A | 0.1 | 80.9 | 81.2 | **82.4** |
| decaf-C → decaf-W | 0.1 | 80.1 | 82.7 | **86.1** | decaf-C → surf-W | 0.1 | 82.0 | 81.3 | **83.5** |
| decaf-C → decaf-D | 0.1 | 80.6 | **83.8** | 83.4 | decaf-C → surf-D | 0.1 | 80.0 | 80.8 | **81.5** |
| decaf-C → decaf-C | 0.2 | **90.6** | 76.7 | 80.5 | decaf-C → surf-C | 0.2 | **66.6** | 63.7 | 65.2 |
| decaf-C → decaf-A | 0.2 | 62.5 | 68.7 | **74.7** | decaf-C → surf-A | 0.2 | 62.9 | 62.4 | **69.3** |
| decaf-C → decaf-W | 0.2 | 65.7 | 75.9 | **79.2** | decaf-C → surf-W | 0.2 | 65.1 | 61.4 | **83.3** |

Table 1: Accuracy for all tasks. The left block are domain adaptation experiments with similar features, where both PGW and UGW are initialised with PW. The right block are domain adaptation experiments with different features, and the reported init is FLB (see Appendix E) used for UGW.

histograms called FLB Mémoli [2011]. We define FLB in our UGW setting as

$$\text{FLB}(\mathcal{X}, \mathcal{Y}) \triangleq \min \int_{X \times Y} |\bar{\mu} \star d_X - \bar{\nu} \star d_Y|^2 \mathrm{d}\pi + \rho \mathrm{KL}(\pi_1|\mu) + \rho \mathrm{KL}(\pi_2|\nu) + \varepsilon \mathrm{KL}(\pi|\mu \otimes \nu), \quad (7)$$

where $\mu \star d_X(x) \triangleq \int d_X(x, x') \mathrm{d}\mu(x')$ is the eccentricity, i.e. a histogram of aggregated distances, and $\bar{\mu} = \mu / m(\mu)$. Contrary to GW Mémoli [2011], there is a priori no link between FLB and UGW.

In the experiments we slightly generalize UGW and use two different marginal penalties $\rho_1 \mathrm{KL}^{\otimes}(\pi_1|\mu) + \rho_2 \mathrm{KL}^{\otimes}(\pi_2|\nu)$ with two parameters $(\rho_1, \rho_2)$ to take into account shifts between domains/features. Note that PGW has a single parameter (which plays a role similar to $(\rho_1, \rho_2)$) which controls the cost of mass creation/destruction. We set $\varepsilon = 2^{-9}$, which avoids introducing an extra parameter in the method. The value $(\rho_1, \rho_2) \in \{2^{-k}, k \in [\![5, 10]\!]\}^2$ are cross validated for each task on the validation folds, and we report the average accuracy on the testing folds. We discuss in Appendix E the impact of reducing the number of parameters on the performance. Comparison with other methods – PU and PUSB Kato et al. [2018], Du Plessis et al. [2014] – are provided in Chapel et al. [2020] and we focus here on the comparison with PGW only.

The results are reported in Table 1. We display the performance of PGW, UGW and the initialization used for UGW to guarantee that using UGW does improve the performance. We observe that when the source and target dataset is the same (C→C tasks), the PW initialization performs better and PGW/UGW degrade the performance, so that in this setting Optimal Transport should be preferred over GW, which is to be expected. However when the domains are different, applying UGW improves the performance over the initialization (which is FLB) in almost all tasks. Note that in that case the methods PU, PUSB or PW cannot be used. Overall, this shows that GW methods are able to solve to some extent the PU learning problem across different spaces, and that using a "softer" KL penalties in UGW is at least at par with Partial GW, and performs better in some settings.

## 5   Conclusion and perspectives

This paper defines two Unbalanced Gromov-Wasserstein formulations: CGW and UGW. We prove that they are both positive and definite. We provide a scalable, GPU-friendly algorithm to compute UGW illustrate its applicability in learning tasks, and show that CGW is a distance between mm-spaces up to isometry. These divergences and distances allow for the first time to blend in a seamless way the transportation geometry of GW with creation and destruction of mass. This hybridization is the key to unlock both theoretical and practical issues. This work opens new questions for future works, for instance removing the bias introduced by the use of entropic regularization, which is important for applications to ML. Note that such a debiasing was successfully applied for Balanced-GW in Bunne et al. [2019] and is shown to lead to a valid divergence for balanced OT in Feydy et al. [2019b] and UW in Séjourné et al. [2019]. The design of efficient numerical solvers for CGW is also an interesting avenue for future works, as well as the study of its induced topology.

## Acknowledgements

The works of Thibault Séjourné and Gabriel Peyré is supported by the ERC grant NORIA. The work of G. Peyré was supported in part by the French government under management of Agence Nationale de la Recherche as part of the "Investissements d'avenir" program, reference ANR19-P3IA-0001 (PRAIRIE 3IA Institute).

The authors thank Rémi Flamary for his remarks and advices, as well as Laetitia Chapel for her help to reproduce her experiments.

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
