# A Background on unbalanced optimal transport

Following Liero et al. [2015], this section reviews and generalizes the homogeneous and conic formulations of unbalanced optimal transport. These three formulations are equal in the convex setting of UOT. Our relaxed divergence UGW and conic distance CGW defined in Section 2 build upon those constructions but are not anymore equal due to the non-convexity of GW problems.

## A.1 Homogeneous formulation

To ease the description of the homogeneous formulation, we develop and refactor the Csiszàr divergence terms of (1) in a form analog to Lemma 1. It reads

$$\mathrm{UW}(\mu,\nu)^q = \inf_{\pi \in \mathcal{M}(X^2)} \int L_{\lambda(d(x,y))}(f(x),g(y))\mathrm{d}\pi(x,y) + \psi'_\infty(|\mu^\perp| + |\nu^\perp|), \tag{8}$$

where $L_c(r,s) \triangleq c + r\varphi(1/r) + s\varphi(1/s)$, $|\mu^\perp| \triangleq \mu^\perp(X)$ and $(f \triangleq \frac{\mathrm{d}\mu}{\mathrm{d}\pi_1}, g \triangleq \frac{\mathrm{d}\nu}{\mathrm{d}\pi_2})$ are the densities of the Lebesgue decomposition of $(\mu,\nu)$ with respect to $(\pi_1, \pi_2)$ and

$$\mu = f\pi_1 + \mu^\perp \quad \text{and} \quad \nu = g\pi_2 + \nu^\perp. \tag{9}$$

Such form is helpful to explicit the terms of pure mass creation/destruction $(|\mu^\perp| + |\nu^\perp|)$ and reinterpret the integral under $\pi$ as a transport term with a new cost $L_{\lambda(d)}$.

Then the authors of Liero et al. [2015] define the homogeneous formulations HUW as

$$\mathrm{HUW}(\mu,\nu)^q \triangleq \inf_{\pi \in \mathcal{M}(X^2)} \int H_{\lambda(d(x,y))}(f(x),g(y))\mathrm{d}\pi(x,y) + \psi'_\infty(|\mu^\perp| + |\nu^\perp|), \tag{10}$$

where the 1-homogeneous function $H_c$ is the perspective transform of $L_c$

$$H_c(r,s) \triangleq \inf_{\theta \geq 0} \theta\big(c + \psi(\tfrac{r}{\theta}) + \psi(\tfrac{s}{\theta})\big) = \inf_{\theta \geq 0} \theta L_c(\tfrac{r}{\theta}, \tfrac{s}{\theta}). \tag{11}$$

By definition one has $L_c \geq H_c$, though after optimization one has UW = HUW.

## A.2 Cone sets, cone distances and explicit settings

The conic formulation detailed in Section A.3 is obtained by performing the optimal transport on the cone set $\mathfrak{C}[X] \triangleq X \times \mathbb{R}_+/(X \times \{0\})$, where the extra coordinate accounts for the mass of the particle. Coordinates of the form $(x,0)$ are merged into a single point called the apex of the cone, noted $\mathfrak{o}_X$. In the sequel, points of $X \times \mathbb{R}_+$ are noted $(x,r)$ and those of $\mathfrak{C}[X]$ are noted $[x,r]$ to emphasize the quotient operation at the apex.

For a pair $(p,q) \in \mathbb{R}_+$, we define for any $[x,r],[y,s] \in \mathfrak{C}[X]^2$

$$\mathcal{D}_{\mathfrak{C}[X]}([x,r],[y,s])^q \triangleq H_{\lambda(d(x,y))}(r^p, s^p). \tag{12}$$

In general $\mathcal{D}_{\mathfrak{C}[X]}$ is not a distance, but it is always definite as proved by the following result described in De Ponti [2019].

**Proposition 5.** *Assume that $d$ is definite, $\lambda^{-1}(\{0\}) = \{0\}$ and $\varphi^{-1}(\{0\}) = \{1\}$. Assume also that for any $(r,s)$, there always exists $\theta^*$ such that $H_c(r,s) = \theta^* L_c(\frac{r}{\theta^*}, \frac{s}{\theta^*})$. Then $\mathcal{D}_{\mathfrak{C}[X]}$ is definite on $\mathfrak{C}[X]$, i.e. $\mathcal{D}_{\mathfrak{C}[X]}([x,r],[y,s]) = 0$ if and only if $(r = s = 0)$ or $(r = s$ and $x = y)$.*

*Proof.* Assume $\mathcal{D}_{\mathfrak{C}[X]}([x,r],[y,s]) = 0$, and write $\theta^*$ such that

$$\mathcal{D}_{\mathfrak{C}[X]}([x,r],[y,s])^q = \theta^* L_c(\tfrac{r^p}{\theta^*}, \tfrac{s^p}{\theta^*}) = \theta^* \lambda(d(x,y)) + r^p\varphi(\tfrac{\theta^*}{r^p}) + s\varphi(\tfrac{\theta^*}{s^p}),$$

where the last line is given by the definition of reverse entropy. There are two cases. If $\theta^* > 0$, since all terms are positive, there are all equal to 0. By definiteness of $d$ it yields $x = y$ and because $\varphi^{-1}(\{0\}) = \{1\}$ we have $r^p = s^p = \theta^*$ and $r = s$. If $\theta^* = 0$ then $\mathcal{D}_{\mathfrak{C}[X]}([x,r],[y,s])^q = \varphi(0)(r^p + s^p)$. The assumption $\varphi^{-1}(\{0\}) = \{1\}$ implies $\varphi(0) > 0$, thus necessarily $r = s = 0$. $\square$

The function $H_c$ can be computed in closed form for a certain number of common entropies $\varphi$, and we refer to Liero et al. [2015, Section 5] for an overview. Of particular interest are those $\varphi$ where $\mathcal{D}_{\mathfrak{C}[X]}$ is a distance, which necessitates a careful choice of $\lambda, p$ and $q$. We now detail three particular settings where this is the case. In each setting we provide $(D_\varphi, \lambda, p, q)$ and its associated cone distance $\mathcal{D}_{\mathfrak{C}[X]}$.

**Gaussian Hellinger distance**    It corresponds to

$$\mathrm{D}_\varphi = \mathrm{KL}, \quad \lambda(t) = t^2 \quad \text{and} \quad q = p = 2,$$
$$\mathcal{D}_{\mathfrak{C}[X]}([x,r],[y,s])^2 = r^2 + s^2 - 2rse^{-d(x,y)/2},$$

in which case it is proved in Liero et al. [2015] that $\mathcal{D}_{\mathfrak{C}[X]}$ is a cone distance.

**Hellinger-Kantorovich / Wasserstein-Fisher-Rao distance**    It reads

$$\mathrm{D}_\varphi = \mathrm{KL}, \quad \lambda(t) = -\log\cos^2(t \wedge \tfrac{\pi}{2}) \quad \text{and} \quad q = p = 2,$$
$$\mathcal{D}_{\mathfrak{C}[X]}([x,r],[y,s])^2 = r^2 + s^2 - 2rs\cos(\tfrac{\pi}{2} \wedge d(x,y)),$$

in which case it is proved in Burago et al. [2001] that $\mathcal{D}_{\mathfrak{C}[X]}$ is a cone distance.

The weight $\lambda(t) = -\log\cos^2(t \wedge \tfrac{\pi}{2})$, which might seem more peculiar, is in fact the penalty that makes unbalanced OT a length space induced by the Gaussian-Hellinger distance (if the ground metric $d$ is itself geodesic), as proved in Liero et al. [2016], Chizat et al. [2018c]. This weight introduces a cut-off, because $\lambda(d(x,y)) = +\infty$ if $d(x,y) > \pi/2$. There is no transport between points too far from each other. The choice of $\pi/2$ is arbitrary, and can be modified by scaling $\lambda \mapsto \lambda(\cdot/s)$ for some cutoff $s$.

**Partial optimal transport**    It corresponds to

$$\mathrm{D}_\varphi = \mathrm{TV}, \quad \lambda(t) = t^q \quad \text{and} \quad q \geq 1 \quad \text{and} \quad p = 1,$$
$$\mathcal{D}_{\mathfrak{C}[X]}([x,r],[y,s])^q = r + s - (r \wedge s)(2 - d(x,y)^q)_+,$$

in which case it is proved in Chizat et al. [2018a] that $\mathcal{D}_{\mathfrak{C}[X]}$ is a cone distance. The case $\mathrm{D}_\varphi = \mathrm{TV}$ is equivalent to partial unbalanced OT, which produces discontinuities (because of the non-smoothness of the divergence) between regions of the supports which are being transported and regions where mass is being destroyed/created. Note that Liero et al. [2015] do not mention that this $\mathcal{D}_{\mathfrak{C}[X]}$ defines a distance, so this result is new to the best of our knowledge, although it can be proved without a conic lifting that partial OT defines a distance as explained in Chizat et al. [2018a].

### A.3   Conic formulation of UW

The last formulation reinterprets UW as an OT problem on the cone, with the addition of two linear constraints. Informally speaking, $H_c$ becomes $\mathcal{D}_{\mathfrak{C}[X]}$, the term $(|\mu^\perp| + |\nu^\perp|)$ is taken into account by the constraints (14) below, and the variables $(f,g)$ are replaced by $(r^p, s^p)$. It reads

$$\mathrm{CUW}(\mu,\nu)^q \triangleq \inf_{\alpha \in \mathcal{U}_p(\mu,\nu)} \int \mathcal{D}_{\mathfrak{C}[X]}([x,r],[y,s]))^q \mathrm{d}\alpha([x,r],[y,s]), \qquad (13)$$

where the constraint set $\mathcal{U}_p(\mu,\nu)$ is defined as

$$\mathcal{U}_p(\mu,\nu) \triangleq \left\{ \alpha \in \mathcal{M}_+(\mathfrak{C}[X]^2) \;:\; \int_{\mathbb{R}_+} r^p \mathrm{d}\alpha_1(\cdot,r) = \mu, \int_{\mathbb{R}_+} s^p \mathrm{d}\alpha_2(\cdot,s) = \nu \right\}. \qquad (14)$$

Thus CUW consists in minimizing the Wasserstein distance $\mathrm{W}_{\mathcal{D}_{\mathfrak{C}[X]}}(\alpha_1,\alpha_2)$ on the cone $(\mathfrak{C}[X], \mathcal{D}_{\mathfrak{C}[X]})$. The additional constraints on $(\alpha_1, \alpha_2)$ mean that the lift of the mass on the cone must be consistent with the total mass of $(\mu,\nu)$. When $\mathcal{D}_{\mathfrak{C}[X]}$ is a distance, CUW inherits the metric properties of $\mathrm{W}_{\mathcal{D}_{\mathfrak{C}[X]}}$. Our theoretical results rely on an analog construction for GW.

The following proposition states the equality of the three formulations and recapitulates its main properties. The proofs are detailed in Liero et al. [2015].

**Proposition 6** (From Liero et al. [2015]). *One has* $\mathrm{UW} = \mathrm{HUW} = \mathrm{CUW}$, *which are symmetric, positive and definite. Furthermore, if* $(X, d_X)$ *and* $(\mathfrak{C}[X], \mathcal{D}_{\mathfrak{C}[X]})$ *are metric spaces with $X$ separable, then* $\mathcal{M}_+(X)$ *endowed with* CUW *is a metric space.*

*Proof.* The equality $\mathrm{UW} = \mathrm{HUW}$ is given by Liero et al. [2015, Theorem 5.8], while the equality $\mathrm{HUW} = \mathrm{CUW}$ holds thanks to Liero et al. [2015, Theorem 6.7 and Remark 7.5], where the latter theorem can be straightforwardly generalized to any cone distance built as in Section 2.2.1. Since

$\mathcal{D}_{\mathfrak{C}[X]}$ is symmetric, positive and definite (see Proposition 3), then so is CUW. Furthermore, if $\mathcal{D}_{\mathfrak{C}[X]}$ satisfies the triangle inequality, separability of $X$ allows to apply the gluing lemma [Liero et al., 2015, Corollary 7.14] which generalizes to any exponent $p$ defining $\mathcal{U}_p(\mu, \nu)$ and any cone distance $\mathcal{D}_{\mathfrak{C}[X]}$. $\qquad\square$

## B  UGW formulation and definiteness

We present in this section the proofs of the properties of our divergence UGW. We refer to Section 2 for the definition of the UGW formulation and its related concepts. For conciseness we write $\Gamma(x, x', y, y') = |d_X(x, x') - d_Y(y, y')|$.

We first start with the existence of minimizers stated in Proposition 1. It illustrates in some sense that our divergence is well-defined.

**Proposition 7** (Existence of minimizers). *Assume $(\mathcal{X}, \mathcal{Y})$ to be compact mm-spaces and that we either have*

1. *$\varphi$ superlinear, i.e $\varphi'_\infty = \infty$*

2. *$\lambda$ has compact sublevel sets in $\mathbb{R}_+$ and $2\varphi'_\infty + \inf \lambda > 0$*

*Then there exists $\pi \in \mathcal{M}_+(X \times Y)$ such that $UGW(\mathcal{X}, \mathcal{Y}) = \mathcal{L}(\pi)$.*

*Proof.* We adapt here from Liero et al. [2015, Theorem 3.3]. The functional is lower semi-continuous as a sum of l.s.c terms. Thus it suffices to have relative compactness of the set of minimizers. Under either one of the assumptions, coercivity of the functional holds thanks to Jensen's inequality

$$\mathcal{L}(\pi) \geq m(\pi)^2 \inf \lambda(\Gamma) + m(\mu)^2 \varphi\left(\frac{m(\pi)^2}{m(\mu)^2}\right) + m(\nu)^2 \varphi\left(\frac{m(\pi)^2}{m(\nu)^2}\right)$$

$$\geq m(\pi)^2 \left[ \inf \lambda(\Gamma) + \frac{m(\mu)^2}{m(\pi)^2} \varphi\left(\frac{m(\pi)^2}{m(\mu)^2}\right) + \frac{m(\nu)^2}{m(\pi)^2} \varphi\left(\frac{m(\pi)^2}{m(\nu)^2}\right) \right].$$

As $m(\pi) \to +\infty$ the right hand side converges to $2\varphi'_\infty + \inf \lambda > 0$, which under either one of the assumptions yields $\mathcal{L}(\pi) \to +\infty$, hence the coercivity. Thus we can assume there exists some $M$ such that $m(\pi) < M$. Since the spaces are assumed to be compact, the Banach-Alaoglu theorem holds and gives relative compactness in $\mathcal{M}_+(X \times Y)$.

Take any sequence of plans $\pi_n$ that approaches $UGW(\mathcal{X}, \mathcal{Y}) = \inf \mathcal{L}(\pi)$. Compactness gives that a subsequence $\pi_{n_k}$ weak* converges to some $\pi^*$. Because $\mathcal{L}$ is l.s.c, we have $\mathcal{L}(\pi^*) \leq \inf \mathcal{L}(\pi)$, thus $\mathcal{L}(\pi^*) = \inf \mathcal{L}(\pi)$. The existence of such limit reaching the infimum gives the existence of a minimizer. $\square$

Note that this formulation is nonegative and symmetric because the functional $\mathcal{L}$ is also nonegative and symmetric in its inputs $(\mathcal{X}, \mathcal{Y})$. This formulation allows straightforwardly to prove the definiteness of UGW.

**Proposition 8** (Definiteness of UGW). *Assume that $\varphi^{-1}(\{0\}) = \{1\}$ and $\lambda^{-1}(\{0\}) = \{0\}$. The following assertions are equivalent:*

1. *$UGW(\mathcal{X}, \mathcal{Y}) = 0$*

2. *$\exists \pi \in \mathcal{M}_+(X \times Y)$ whose marginals are $(\mu, \nu)$ such that $d_X(x, x') = d_Y(y, y')$ for $\pi \otimes \pi$-a.e. $(x, x', y, y') \in (X \times Y)^2$.*

3. *There exists a mm-space $(Z, d_Z, \eta)$ with full support and Borel maps $\psi_X : Z \to X$ and $\psi_Y : Z \to Y$. such that $(\psi_X)_\sharp \eta = \mu$, $(\psi_Y)_\sharp \eta = \nu$ and $d_Z = (\psi_X)^\sharp d_X = (\psi_Y)^\sharp d_Y$*

4. *There exists a Borel measurable bijection between the measures' supports $\psi : spt(\mu) \to spt(\nu)$ with Borel measurable inverse such that $\psi_\sharp \mu = \nu$ and $d_Y = \psi^\sharp d_X$.*

*Proof.* Recall that $(2) \Leftrightarrow (3) \Leftrightarrow (4)$ from Sturm [2012, Lemma 1.10]. thus it remains to prove $(1) \Leftrightarrow (2)$.

If there is such coupling plan $\pi$ between $(\mu, \nu)$ then one has $\pi \otimes \pi$-a.e. that $\Gamma = 0$, and all $\varphi$-divergences are zero as well, yielding a distance of zero a.e.

Assume now that $UGW(\mathcal{X}, \mathcal{Y}) = 0$, and write $\pi$ an optimal plan. All terms of $\mathcal{L}$ are positive, thus under our assumptions we have $\Gamma = 0$, $\pi_1 \otimes \pi_1 = \mu \otimes \mu$ and $\pi_2 \otimes \pi_2 = \nu \otimes \nu$. Thus we get that $\pi$ has marginals $(\mu, \nu)$ and that $d_X(x, x') = d_Y(y, y')$ $\pi \otimes \pi$-a.e. $\square$

We end with a result on the reformulation of UGW which is the first step to connnect it with the conic formulation CGW. It is the same proof as in the main body.

**Lemma 2.** *Defining* $L_c(r,s) \triangleq c + r\varphi(1/r) + s\varphi(1/s)$, *and writing* $(f \triangleq \frac{\mathrm{d}\mu}{\mathrm{d}\pi_1}, g \triangleq \frac{\mathrm{d}\nu}{\mathrm{d}\pi_2})$ *the Lebesgue densities of* $(\mu,\nu)$ *w.r.t.* $(\pi_1, \pi_2)$ *such that* $\mu = f\pi_1 + \mu^\perp$ *and* $\nu = g\pi_2 + \nu^\perp$, *one has*

$$\mathcal{L}(\pi) = \int_{X^2 \times Y^2} L_{\lambda(\Gamma)}(f \otimes f, g \otimes g)\mathrm{d}\pi\mathrm{d}\pi + \varphi(0)(|(\mu \otimes \mu)^\perp| + |(\nu \otimes \nu)^\perp|).$$

*Proof.* Using Equation (24), one has

$$\begin{aligned}
\mathcal{L}(\pi) &= \int_{X^2 \times Y^2} \lambda(\Gamma)\mathrm{d}\pi\mathrm{d}\pi + \mathrm{D}_\varphi^\otimes(\pi_1|\mu) + \mathrm{D}_\varphi^\otimes(\pi_2|\nu) \\
&= \int_{X^2 \times Y^2} \lambda(\Gamma)\mathrm{d}\pi\mathrm{d}\pi + \mathrm{D}_\psi^\otimes(\mu|\pi_1) + \mathrm{D}_\psi^\otimes(\nu|\pi_2) \\
&= \int_{X^2 \times Y^2} \lambda(\Gamma)\mathrm{d}\pi\mathrm{d}\pi + \int_{X^2} \psi(f \otimes f)\mathrm{d}\pi_1\mathrm{d}\pi_1 + \int_{Y^2} \psi(g \otimes g)\mathrm{d}\pi_2\mathrm{d}\pi_2 \\
&\quad + \varphi(0)(|(\mu \otimes \mu)^\perp| + |(\nu \otimes \nu)^\perp|) \\
&= \int_{X^2 \times Y^2} L_{\lambda(\Gamma)}(f \otimes f, g \otimes g)\mathrm{d}\pi\mathrm{d}\pi + \varphi(0)(|(\mu \otimes \mu)^\perp| + |(\nu \otimes \nu)^\perp|).
\end{aligned}$$

$\square$

## C   Conic formulation and metric properties

We present in this section the proofs of the properties mentioned in Section 2. We refer to Section 2 and Appendix A for the definition of the conic formulation and its related concepts.

In this section we frequently use the notion of marginal for neasures. For any sets $E, F$, we write $\mathfrak{P}^{(E)} : E \times F \to E$ the **canonical projection** such that for any $(x,y) \in E \times F$, $\mathfrak{P}^{(E)}(x,y) = x$. Consider two complete separable mm-spaces $\mathcal{X} = (X, d_X, \mu)$ and $\mathcal{Y} = (Y, d_Y, \nu)$. Write $\pi \in \mathcal{M}_+(X \times Y)$ a coupling plan, and define its marginals by $\pi_1 = \mathfrak{P}_\sharp^{(X)}\pi$ and $\pi_2 = \mathfrak{P}_\sharp^{(Y)}\pi$. The definition of the marginals can also be seen by the use of test functions. In the case of $\pi_1$ it reads for any test function $\xi$

$$\int \xi(x)\mathrm{d}\pi_1(x) = \int \xi(x)\mathrm{d}\pi(x, y).$$

### C.1   Preliminary results

We present in this section concepts and properties which are necessary for the proof of Theorem 1. We introduce a dilation operator whose role is to rescale the radial coordinate of a measure with a given scaling.

**Definition 2** (dilations). *Consider* $v([x,r], [y,s])$ *a Borel measurable scaling function depending on* $[x,r], [y,s] \in \mathfrak{C}[X] \times \mathfrak{C}[Y]$. *Take a plan* $\alpha \in \mathcal{M}_+(\mathfrak{C}[X] \times \mathfrak{C}[Y])$. *We define the dilation* $\mathrm{Dil}_v : \alpha \mapsto (h_v)_\sharp(v^p\alpha)$ *where*

$$h_v([x,r], [y,s]) \triangleq ([x, r/w], [y, s/w]),$$

*where* $w = v([x,r], [y,s])$. *It reads for any test function* $\xi$

$$\int \xi([x,r], [y,s])\mathrm{dDil}_v(\alpha) = \int \xi([x, r/w], [y, s/w])w^p\mathrm{d}\alpha.$$

The importance of dilations is given by the following lemma.

**Lemma 3** (Invariance to dilation). *The problem* CGW *is invariant to dilations, i.e. for any* $\alpha \in \mathcal{U}_p(\mu, \nu)$, *we have* $\mathrm{Dil}_v(\alpha) \in \mathcal{U}_p(\mu, \nu)$ *and* $\mathcal{H}(\alpha) = \mathcal{H}(\mathrm{Dil}_v(\alpha))$.

*Proof.* First we prove the stability of $\mathcal{U}_p(\mu, \nu)$ under dilations. Take $\alpha \in \mathcal{U}_p(\mu, \nu)$. For any test function $\xi$ defined on $X$ we have

$$\int \xi(x) r^p \mathrm{dDil}_v(\alpha) = \int \xi(x) (\frac{r}{v})^p . v^p \mathrm{d}(\alpha) = \int \xi(x) r^p \mathrm{d}\alpha = \int \xi(x) \mathrm{d}\mu(x).$$

Similarly we get $\mathfrak{P}_\sharp^{(Y)}(s^q \mathrm{Dil}_v(\alpha)) = \nu$, thus $\mathrm{Dil}_v(\alpha) \in \mathcal{U}_p(\mu, \nu)$.

It remains to prove the invariance of the functional. Recall that $\mathcal{D}^q$ is p-homogeneous. It yields

$$\mathcal{H}(\mathrm{Dil}_v(\alpha)) = \int \mathcal{D}([d_X(x, x'), rr'], [d_Y(y, y'), ss'])^q \mathrm{dDil}_v(\alpha) \mathrm{dDil}_v(\alpha)$$

$$= \int \mathcal{D}([d_X(x, x'), \frac{r}{v} \cdot \frac{r'}{v}], [d_Y(y, y'), \frac{s}{v} \cdot \frac{s'}{v}])^q v^p \cdot v^p \mathrm{d}\alpha \mathrm{d}\alpha$$

$$= \int \frac{1}{v^{2p}} \mathcal{D}([d_X(x, x'), rr'], [d_Y(y, y'), ss'])^q v^{2p} \mathrm{d}\alpha \mathrm{d}\alpha$$

$$= \int \mathcal{D}([d_X(x, x'), rr'], [d_Y(y, y'), ss'])^q \mathrm{d}\alpha \mathrm{d}\alpha$$

$$= \mathcal{H}(\alpha)$$

Both the functional and the constraint set are invariant, thus the whole CGW problem is invariant to dilations. $\square$

The above lemma allows to normalize the plan such that one of its marginal is fixed to some value. Fixing a marginal allows to generalize the gluing lemma which is a key ingredient of the triangle inequality in optimal transport.

**Lemma 4** (Normalization lemma)**.** *Assume there exists $\alpha \in \mathcal{U}_p(\mu, \nu)$ such that $\mathrm{CGW}(\mathcal{X}, \mathcal{Y}) = \mathcal{H}(\alpha)$. Then there exists $\tilde{\alpha}$ such that $\tilde{\alpha} \in \mathcal{U}_p(\mu, \nu)$ and $\mathrm{CGW}(\mathcal{X}, \mathcal{Y}) = \mathcal{H}(\tilde{\alpha})$ and whose marginal on $\mathfrak{C}[Y]$ is $\nu_{\mathfrak{C}[Y]} = \mathfrak{P}^{(\mathfrak{C}[Y])} \sharp \tilde{\alpha} = \delta_{\mathfrak{o}_Y} + \mathfrak{p}_\sharp(\nu \otimes \delta_1)$, where $\mathfrak{p}$ is the canonical injection from $Y \times \mathbb{R}_+$ to $\mathfrak{C}[Y]$.*

*Proof.* The proof is exactly the same as Liero et al. [2015, Lemma 7.10] and is included for completeness. Take an optimal plan $\alpha$. Because the functional and the constraints are homogeneous in $(r, s)$, the plan $\hat{\alpha} = \alpha + \delta_{\mathfrak{o}_X} \otimes \delta_{\mathfrak{o}_Y}$ verifies $\hat{\alpha} \in \mathcal{U}_p(\mu, \nu)$ and $\mathcal{H}(\hat{\alpha}) = \mathcal{H}(\alpha)$. Indeed, because of this homogeneity the contribution $\delta_{\mathfrak{o}_X} \otimes \delta_{\mathfrak{o}_Y}$ has $(r, s) = (0, 0)$ which has thus no impact.

Considering $\hat{\alpha}$ instead of $\alpha$ allows to assume without loss of generality that the transport plan charges the apex, i.e. setting

$$S = \{[x, r], [y, s] \in \mathfrak{C}[X] \times \mathfrak{C}[Y], [y, s] = \mathfrak{o}_Y\}, \tag{15}$$

one has $\omega_Y \triangleq \hat{\alpha}(S) \geq 1$. Then we can define the following scaling

$$v([x, r], [y, s]) = \begin{cases} s \text{ if } s > 0 \\ \omega_Y^{-1/q} \text{ otherwise.} \end{cases} \tag{16}$$

We prove now that $\mathrm{Dil}_v(\hat{\alpha})$ has the desired marginal on $\mathfrak{C}(Y)$ by considering test functions $\xi([y, s])$. We separate the integral into two parts with the set $S$, and write $\hat{\alpha} = \hat{\alpha}|_S + \hat{\alpha}|_{S^c}$ their restrictions to $S$ and $S^c$ respectively. It reads

$$\int \xi([y, s]) \mathrm{dDil}_v(\hat{\alpha}) = \int \xi([y, s/v]) v^p \mathrm{d}\hat{\alpha}$$

$$= \int \xi([y, s/v]) v^p \mathrm{d} \hat{\alpha}|_S + \int \xi([y, s/v]) v^p \mathrm{d} \hat{\alpha}|_{S^c}$$

$$= \int \xi(\mathfrak{o}_Y) \omega_Y^{-1} \mathrm{d} \hat{\alpha}|_S + \int \xi([y, s/s]) s^p \mathrm{d} \hat{\alpha}|_{S^c}$$

$$= \xi(\mathfrak{o}_Y) \cdot \omega_Y \cdot \omega_Y^{-1} + \int \xi([y, 1]) s^p \mathrm{d}\hat{\alpha}$$

$$= \xi(\mathfrak{o}_Y) + \int \xi(\mathfrak{p}(y, s)) \mathrm{d}(\nu(y) \otimes \delta_1(s))$$

$$= \int \xi([y, s]) \mathrm{d}(\delta_{\mathfrak{o}_Y} + \mathfrak{p}_\sharp(\nu \otimes \delta_1)),$$

which is the formula of the desired marginal on $\mathfrak{C}[Y]$. Since $\hat{\alpha} \in \mathcal{U}_p(\mu, \nu)$, its dilation is also in $\mathcal{U}_p(\mu, \nu)$, and $\mathcal{H}(\alpha) = \mathcal{H}(\hat{\alpha}) = \mathcal{H}(\mathrm{Dil}_v(\hat{\alpha}))$. $\qquad\qquad\square$

### C.1.1 Proof of Theorem 1

*Non-negativity* and *symmetry* hold since $\mathcal{H}$ is a sum of non-negative symmetric terms. To prove *Definiteness*, assume $\mathrm{CGW}(\mathcal{X}, \mathcal{Y}) = 0$, and write $\alpha$ an optimal plan. We have $\alpha \otimes \alpha$-a.e. that $d_X(x, x') = d_Y(y, y')$ and $rr' = ss'$ because $\mathcal{D}$ is definite (see Proposition 3). Thanks to the completeness of $(\mathcal{X}, \mathcal{Y})$ and a result from Sturm [2012, Lemma 1.10], such property implies the existence of a Borel isometric bijection with Borel inverse between the supports of the measures $\psi : \mathrm{Supp}(\mu) \to \mathrm{Supp}(\nu)$, where Supp denotes the support. The bijection $\psi$ verifies $d_X(x, x') = d_Y(\psi(x), \psi(x'))$. To prove $\mathcal{X} \sim \mathcal{Y}$ it remains to prove $\psi_\sharp \mu = \nu$. Due to the density of continuous functions of the form $\xi(x)\xi(x')$, the constraints of $\mathcal{U}_p(\mu, \nu)$ are equivalent to

$$\int_{\mathbb{R}_+} (rr')^p \mathrm{d}\alpha_1(\cdot, r) \mathrm{d}\alpha_1(\cdot, r') = \mu \otimes \mu, \quad \int_{\mathbb{R}_+} (ss')^p \mathrm{d}\alpha_2(\cdot, s) \mathrm{d}\alpha_2(\cdot, s') = \nu \otimes \nu.$$

Take a continuous test function $\xi$ defined on $\mathrm{Supp}(\nu)^2$. Writing $y = \psi(x)$ and $y' = \psi(x')$, one has

$$\int \xi(y, y') \mathrm{d}\nu \mathrm{d}\nu = \int \xi(y, y')(ss')^p \mathrm{d}\alpha \mathrm{d}\alpha$$

$$= \int \xi(\psi(x), \psi(x'))(ss')^p \mathrm{d}\alpha \mathrm{d}\alpha$$

$$= \int \xi(\psi(x), \psi(x'))(rr')^p \mathrm{d}\alpha \mathrm{d}\alpha$$

$$= \int \xi(\psi(x), \psi(x')) \mathrm{d}\mu \mathrm{d}\mu$$

$$= \int \tilde{\xi}(x, x') \mathrm{d}\psi_\sharp \mu \mathrm{d}\psi_\sharp \mu.$$

Since $\psi$ is a bijection, there is a bijection between continuous functions $\xi$ of $\mathrm{Supp}(\nu)^2$ and functions $\tilde{\xi}$ of $\mathrm{Supp}(\mu)^2$. Thus we obtain $\nu = \psi_\sharp \mu$ and we have $\mathcal{X} \sim \mathcal{Y}$.

It remains to prove the *triangle inequality*. Assume now that $\mathcal{D}$ satisfies it. Given three mm-spaces $(\mathcal{X}, \mathcal{Y}, \mathcal{Z})$ respectively equipped with measures $(\mu, \nu, \eta)$, consider $\alpha, \beta$ which are optimal plans for $\mathrm{CGW}(\mathcal{X}, \mathcal{Y})$ and $\mathrm{CGW}(\mathcal{Y}, \mathcal{Z})$. Using Lemma 4 to both $\alpha$ and $\beta$, we can consider measures $(\bar{\alpha}, \bar{\beta})$ which are also optimal and have a common marginal $\bar{\nu}$ on $\mathfrak{C}[Y]$. Thanks to this common marginal and the separability of $(X, Y, Z)$, the standard gluing lemma [Villani, 2003, Lemma 7.6] applies and yields a glued plan $\gamma \in \mathcal{M}_+(\mathfrak{C}[X] \times \mathfrak{C}[Y] \times \mathfrak{C}[Z])$ whose respective marginals on $\mathfrak{C}[X] \times \mathfrak{C}[Y]$ and $\mathfrak{C}[Y] \times \mathfrak{C}[Z]$ are $(\bar{\alpha}, \bar{\beta})$. Furthermore, the marginal $\bar{\gamma}$ of $\gamma$ on $\mathfrak{C}[X] \times \mathfrak{C}[Z]$ is in $\mathcal{U}_p(\mu, \eta)$. Indeed, $(\bar{\gamma}, \bar{\alpha})$ have the same marginal on $\mathfrak{C}[X]$ and same for $(\bar{\gamma}, \bar{\beta})$ on $\mathfrak{C}[Z]$, hence this property. Write

$d_X = d_X(x, x')$ for sake of conciseness (and similarly for $Y, Z$). The calculation reads

$$\text{CGW}(\mathcal{X}, \mathcal{Z})^{\frac{1}{q}} \tag{17}$$

$$\leq \left( \int \mathcal{D}([d_X, rr'], [d_Z, tt'])^q \mathrm{d}\bar{\gamma}([x, r], [z, t]) \mathrm{d}\bar{\gamma}([x', r'], [z', t']) \right)^{\frac{1}{q}} \tag{18}$$

$$\leq \left( \int \mathcal{D}([d_X, rr'], [d_Z, tt'])^q \mathrm{d}\gamma([x, r], [y, s], [z, t]) \mathrm{d}\gamma([x', r'], [y', s'], [z', t']) \right)^{\frac{1}{q}} \tag{19}$$

$$\leq \left( \int (\mathcal{D}([d_X, rr'], [d_Y, ss']) + \mathcal{D}([d_Y, ss'], [d_Z, tt']))^q \mathrm{d}\gamma \mathrm{d}\gamma \right)^{\frac{1}{q}} \tag{20}$$

$$\leq \left( \int \mathcal{D}([d_X, rr'], [d_Y, ss'])^q \mathrm{d}\gamma \mathrm{d}\gamma \right)^{\frac{1}{q}} + \left( \int \mathcal{D}([d_Y, ss'], [d_Z, tt'])^q \mathrm{d}\gamma \mathrm{d}\gamma \right)^{\frac{1}{q}} \tag{21}$$

$$\leq \left( \int \mathcal{D}([d_X, rr'], [d_Y, ss'])^q \mathrm{d}\bar{\alpha}([x, r], [y, s]) \mathrm{d}\bar{\alpha}([x', r'], [y', s']) \right)^{\frac{1}{q}}$$
$$+ \left( \int \mathcal{D}([d_Y, ss'], [d_Z, tt'])^q \mathrm{d}\bar{\beta}([y, s], [z, t]) \mathrm{d}\bar{\beta}([y', s'], [z', t']) \right)^{\frac{1}{q}} \tag{22}$$

$$\leq \text{CGW}(\mathcal{X}, \mathcal{Y})^{\frac{1}{q}} + \text{CGW}(\mathcal{Y}, \mathcal{Z})^{\frac{1}{q}}. \tag{23}$$

Since $\bar{\gamma} \in \mathcal{U}_p(\mu, \eta)$, it is thus suboptimal, which yields Equation (18). Because $\bar{\gamma}$ is the marginal of $\gamma$ we get Equation (19). Equations (20) and (21) are respectively obtained by the triangle and Minkowski inequalities, which hold because $\mathcal{D}$ which is a distance. Equation (22) is the marginalization of $\gamma$, and Equation (23) is given by the optimality of $(\bar{\alpha}, \bar{\beta})$, which ends the proof of the triangle inequality.

### C.1.2 Proof of the inequality between UGW and CGW

The proof consists in considering an optimal plan $\pi$ for UGW, building a lift $\alpha$ of this plan into the cone such that $\mathcal{L}(\pi) \geq \mathcal{H}(\alpha)$, and prove that $\alpha$ is admissible for the program CGW, thus suboptimal.

Using Equation (9), we have

$$\mu \otimes \mu = (f \otimes f)\pi_1 \otimes \pi_1 + (\mu \otimes \mu)^\perp,$$
$$(\mu \otimes \mu)^\perp = \mu^\perp \otimes (f\pi_1) + (f\pi_1) \otimes \mu^\perp + \mu^\perp \otimes \mu^\perp,$$
$$\nu \otimes \nu = (g \otimes g)\pi_2 \otimes \pi_2 + (\nu \otimes \nu)^\perp, \tag{24}$$
$$(\nu \otimes \nu)^\perp = \nu^\perp \otimes (g\pi_2) + (g\pi_2) \otimes \nu^\perp + \nu^\perp \otimes \nu^\perp.$$

Recall that the canonic injection $\mathfrak{p}$ reads $\mathfrak{p}(x, r) = [x, r]$. Based on the above Lebesgue decomposition, we define the conic plan

$$\alpha = (\mathfrak{p}(x, f(x)^{\frac{1}{p}}), \mathfrak{p}(y, g(y)^{\frac{1}{p}}))_\sharp \pi(x, y) + \delta_{\mathfrak{o}_X} \otimes \mathfrak{p}_\sharp [\nu^\perp \otimes \delta_1] + \mathfrak{p}_\sharp [\mu^\perp \otimes \delta_1] \otimes \delta_{\mathfrak{o}_Y}. \tag{25}$$

We have that $\alpha \in \mathcal{U}_p(\mu, \nu)$. Indeed for the first marginal (and similarly for the second) we have for any test function $\xi(x)$

$$\int \xi(x)(r)^p \mathrm{d}\alpha = \int \xi(x)f(x)\mathrm{d}\pi_1(x) + 0 + \int \xi(x)(1)^p \mathrm{d}\mu^\perp(x)$$
$$= \int \xi(x)\mathrm{d}(f(x)\pi_1 + \mu^\perp)$$
$$= \int \xi(x)\mathrm{d}\mu(x).$$

We define $\theta^* = \theta_c^*(r, s)$ the parameter which verifies $H_c(r, s) = \theta^* L_c(r/\theta^*, s/\theta^*)$. We restrict $\alpha \otimes \alpha$ to the set $S = \{\theta_{\lambda(\Gamma)}^*((rr')^p, (ss')^p) > 0\}$. By construction, $\theta_c^*(r, s)$ is 1-homogeneous in $(r, s)$. Thus on S we necessarily have $r, r', s, s' > 0$. It yields

$$\alpha \otimes \alpha|_S = (\mathfrak{p}(x, f(x)^{\frac{1}{p}}), \mathfrak{p}(y, g(y)^{\frac{1}{p}}), \mathfrak{p}(x', f(x')^{\frac{1}{p}}), \mathfrak{p}(y', g(y')^{\frac{1}{p}}))_\sharp (\pi \otimes \pi).$$

Concerning the orthogonal part of the decomposition, note that whenever $\theta^* = 0$, due to the definition of $H$ the cone distance reads

$$\mathcal{D}([x,r],[y,s])^q = \varphi(0)(r^p + s^p). \tag{26}$$

It geometrically means that the shortest path between $[x,r]$ and $[y,s]$ must pass via the apex, which corresponds to a pure mass creation/destruction regime.

Furthermore we have that

$$|(\mu \otimes \mu)^\perp| = \int (r \cdot r')^p \mathrm{d}\,(\alpha \otimes \alpha)|_{S^c},$$

$$|(\nu \otimes \nu)^\perp| = \int (s \cdot s')^p \mathrm{d}\,(\alpha \otimes \alpha)|_{S^c}.$$

Indeed, thanks to Equation (25) we have for the first marginal that

$$
\begin{aligned}
|(\mu \otimes \mu)^\perp| &= \left(\mu^\perp \otimes (f\pi_1) + (f\pi_1) \otimes \mu^\perp + \mu^\perp \otimes \mu^\perp\right)(X^2)\\
&= \int (rr')^p \mathrm{d}\mathfrak{p}_\sharp[\mu^\perp \otimes \delta_1] \mathrm{d}\mathfrak{p}(x', f(x')^{\frac{1}{p}})_\sharp \pi_1(x')\\
&\quad + \int (rr')^p \mathrm{d}\mathfrak{p}(x, f(x)^{\frac{1}{p}})_\sharp \pi_1(x) \mathrm{d}\mathfrak{p}_\sharp[\mu^\perp \otimes \delta_1]\\
&\quad + \int (rr')^p \mathrm{d}\mathfrak{p}_\sharp[\mu^\perp \otimes \delta_1] \mathrm{d}\mathfrak{p}_\sharp[\mu^\perp \otimes \delta_1]\\
&= \int (rr')^p \mathrm{d}\,(\alpha \otimes \alpha)|_{S^c}.
\end{aligned}
$$

Note that the last equality holds because each term of $\alpha \otimes \alpha$ involving a measure $\delta_{\mathfrak{o}_X}$ cancels out when integrated against $(rr')^p$.

Eventually the computation gives (thanks to Lemma 1)

$$
\begin{aligned}
\mathcal{L}(\pi) &= \int_{X^2 \times Y^2} L_{\lambda(\Gamma)}(f \otimes f, g \otimes g) \mathrm{d}\pi \mathrm{d}\pi + \varphi(0)(|(\mu \otimes \mu)^\perp| + |(\nu \otimes \nu)^\perp|)\\
&\geq \int H_{\lambda(\Gamma)}(f \otimes f, g \otimes g) \mathrm{d}\pi \mathrm{d}\pi + \varphi(0)(|(\mu \otimes \mu)^\perp| + |(\nu \otimes \nu)^\perp|)\\
&\geq \int \mathcal{D}([d_X(x,x'),(f \otimes f)^{\frac{1}{p}}],[d_Y(y,y'),(g \otimes g)^{\frac{1}{p}}])^q \mathrm{d}\pi \mathrm{d}\pi\\
&\quad + \int \varphi(0)(rr')^p \mathrm{d}\,(\alpha \otimes \alpha)|_{S^c} + \int \varphi(0)(ss')^p \mathrm{d}\,(\alpha \otimes \alpha)|_{S^c}\\
&\geq \int \mathcal{D}([d_X(x,x'),rr'],[d_Y(y,y'),ss'])^q \mathrm{d}\,(\alpha \otimes \alpha)|_S\\
&\quad + \int \varphi(0)((rr')^p + (ss')^p) \mathrm{d}\,(\alpha \otimes \alpha)|_{S^c}\\
&\geq \int \mathcal{D}([d_X(x,x'),rr'],[d_Y(y,y'),ss'])^q \mathrm{d}\alpha \mathrm{d}\alpha\\
&\geq \mathcal{H}(\alpha).
\end{aligned}
$$

Thus we have $\mathrm{UGW}(\mathcal{X},\mathcal{Y}) = \mathcal{L}(\pi) \geq \mathcal{H}(\alpha) \geq \mathrm{CGW}(\mathcal{X},\mathcal{Y})$.

# D   Optimization, algorithms and formulas

We present in this section the important results of Section 3. We start with Theorem 2 stating that for a wide range of quadratic programs, performing a bi-convex relaxation yields the same objective value as the original program. We prove its application in Theorem 3. We provide a decomposition property of $\mathrm{KL}^{\otimes}$, followed by the proof of Proposition 4, and a description of the algorithm in a discrete setting, where computationnaly implementable formulas are provided.

## D.1   Proof of Theorem 2

*Proof.* The function $\mathcal{F}$ is the symmetrization of $\mathcal{L}$, so that $\mathcal{F}(\pi, \pi) = \mathcal{L}(\pi)$. By the hypothesis on $\mathcal{L}$, the mimimum values of the functions (if it exists) are finite. The two following inequalities are obtained by optimality of $(\pi_*, \gamma_*)$,

$$
\begin{cases}
\mathcal{F}(\pi_*, \gamma_*) \leq \mathcal{F}(\pi_*, \pi_*) \\
\mathcal{F}(\pi_*, \gamma_*) \leq \mathcal{F}(\gamma_*, \gamma_*) \,.
\end{cases}
\tag{27}
$$

Note that the hypotheses imply that $\mathcal{F}(\pi_*, \pi_*)$ and $\mathcal{F}(\gamma_*, \gamma_*)$ are both finite. Combining these two inequalities leads to $\mathcal{F}(\pi_*, \pi_*) + \mathcal{F}(\gamma_*, \gamma_*) - 2\mathcal{F}(\pi_*, \gamma_*) \geq 0$, which implies

$$
\frac{1}{2}\langle \pi_* - \gamma_*, k(\pi_* - \gamma_*) \rangle \geq 0 \,,
\tag{28}
$$

since the separable parts in $\mathcal{F}$ cancel. Since $k$ is negative, we also have the converse inequality, thus $\frac{1}{2}\langle \pi_* - \gamma_*, k(\pi_* - \gamma_*) \rangle = 0$. Therefore, we deduce when $k$ is definite that $\pi_* = \gamma_*$.

We now treat the case when $k$ is not definite. In this case, we only have $\frac{1}{2}\langle \pi_* - \gamma_*, k(\pi_* - \gamma_*) \rangle = 0$ which implies that $\pi_* - \gamma_* \in \mathrm{Ker}(k)$ since $k$ is non positive. The first inequality in (27) implies $f(\pi_*) \leq f(\gamma_*)$ and by symmetry $f(\pi_*) = f(\gamma_*)$ and as a conclusion $\mathcal{F}(\pi_*, \pi_*) = \mathcal{F}(\pi_*, \gamma_*) = \mathcal{F}(\gamma_*, \gamma_*)$.

The last case follows from the observation that on the segment $[\pi_*, \gamma_*] \subset C$, the quadratic part of $\mathcal{F}$ is constant. Indeed one has for $t \in [0, 1]$, for $z = t(\pi_* - \gamma_*) + \gamma_*$ one has

$$
\langle z, k(z) \rangle = t^2 \langle (\pi_* - \gamma_*), k(\pi_* - \gamma_*) \rangle + 2t \langle \gamma_*, k(\pi_* - \gamma_*) \rangle + \langle \gamma_*, k(\gamma_*) \rangle = \langle \gamma_*, k(\gamma_*) \rangle,
$$

since $\pi_* - \gamma_* \in \mathrm{Ker}(k)$. Thus minimizing $\mathcal{F}$ on $[\pi_*, \gamma_*]$ is reduced to the minimization of $f$ on this segment. By the above remark, $f(\pi_*) = f(\gamma_*)$ which implies $\pi_* = \gamma_*$ by strict convexity.    $\square$

## D.2   Properties of the quadratic KL divergence

We present in this section an additional property on the quadratic-KL divergence which allows to reduce the computational burden to evaluate it by involving the computation of a standard KL divergence.

**Proposition 9.** *For any measures* $(\mu, \nu) \in \mathcal{M}_+(\mathcal{X})$, *one has*

$$
\begin{aligned}
\mathrm{KL}(\mu \otimes \nu | \alpha \otimes \beta) = {} & m(\nu)\mathrm{KL}(\mu|\alpha) + m(\mu)\mathrm{KL}(\nu|\beta) \\
& + (m(\mu) - m(\alpha))(m(\nu) - m(\beta)).
\end{aligned}
\tag{29}
$$

*In particular,*

$$
\mathrm{KL}(\mu \otimes \mu | \nu \otimes \nu) = 2m(\mu)\mathrm{KL}(\mu|\nu) + (m(\mu) - m(\nu))^2.
\tag{30}
$$

*Proof.* Assuming $\mathrm{KL}(\mu \otimes \nu | \alpha \otimes \beta)$ to be finite, one has $\mu = f\alpha$ and $\nu = g\beta$. It reads

$$\mathrm{KL}(\mu \otimes \nu | \alpha \otimes \beta) = \int \log(f \otimes g) \mathrm{d}\mu \mathrm{d}\nu - m(\mu)m(\nu) + m(\alpha)m(\beta)$$

$$= m(\nu) \int \log(f) \mathrm{d}\mu + m(\mu) \int \log(g) \mathrm{d}\nu$$
$$- m(\mu)m(\nu) + m(\alpha)m(\beta)$$
$$= m(\nu)\big[\mathrm{KL}(\mu|\alpha) + m(\mu) - m(\alpha)\big]$$
$$+ m(\mu)\big[\mathrm{KL}(\nu|\beta) + m(\nu) - m(\beta)\big]$$
$$- m(\mu)m(\nu) + m(\alpha)m(\beta)$$
$$= m(\nu)\mathrm{KL}(\mu|\alpha) + m(\mu)\mathrm{KL}(\nu|\beta)$$
$$+ m(\mu)m(\nu) - m(\nu)m(\alpha) - m(\mu)m(\beta) + m(\alpha)m(\beta)$$
$$= m(\nu)\mathrm{KL}(\mu|\alpha) + m(\mu)\mathrm{KL}(\nu|\beta)$$
$$+ (m(\mu) - m(\alpha))(m(\nu) - m(\beta)).$$

$\square$

In the Balanced setting, with $(\mu, \nu)$ probabilities, the regularization reads $\mathrm{KL}^{\otimes}(\pi | \mu \otimes \nu) = 2\mathrm{KL}(\pi | \mu \otimes \nu)$. Thus (up to a factor 2) we retrieve as a particular case the setting of Peyré et al. [2016].

### D.3  Proof of Proposition 4

We now prove Proposition 4 which applies the above result.

**Proposition 10.** *For a fixed $\gamma$, the optimal $\pi \in \arg\min_{\pi} \mathcal{F}(\pi, \gamma) + \varepsilon\mathrm{KL}(\pi \otimes \gamma | (\mu \otimes \nu)^{\otimes 2})$ is the solution of $\min_{\pi} \int c_{\gamma}^{\varepsilon}(x, y) \mathrm{d}\pi(x, y) + \rho m(\gamma)\mathrm{KL}(\pi_1|\mu) + \rho m(\gamma)\mathrm{KL}(\pi_2|\nu) + \varepsilon m(\gamma)\mathrm{KL}(\pi|\mu \otimes \nu)$, where $m(\gamma) \triangleq \gamma(X \times Y)$ is the total mass of $\gamma$, and where we define the cost and weight associated to $\gamma$ as*

$$c_{\gamma}^{\varepsilon}(x, y) \triangleq \int \lambda(\Gamma(x, \cdot, y, \cdot)) \mathrm{d}\gamma + \rho \int \log(\frac{\mathrm{d}\gamma_1}{\mathrm{d}\mu}) \mathrm{d}\gamma_1 + \rho \int \log(\frac{\mathrm{d}\gamma_2}{\mathrm{d}\nu}) \mathrm{d}\gamma_2 + \varepsilon \int \log(\frac{\mathrm{d}\gamma}{\mathrm{d}\mu \mathrm{d}\nu}) \mathrm{d}\gamma.$$

*Proof.* First note that $\mathcal{F}(\gamma, \pi) = \mathcal{F}(\pi, \gamma)$ so that minimizing with the first or the second argument gives the same solution. Setting $\gamma$ to be fixed, the rest follows from the factorisation

$$\mathrm{KL}(\pi_1 \otimes \gamma_1 | \mu \otimes \mu) = m(\gamma)\mathrm{KL}(\pi_1|\mu) + m(\pi)\mathrm{KL}(\gamma_1|\mu) + (m(\gamma) - m(\mu))(m(\pi) - m(\mu))$$

$$= m(\pi)\big[\mathrm{KL}(\gamma_1|\mu) + m(\gamma) - m(\mu)\big] + m(\gamma)\mathrm{KL}(\pi_1|\mu) - m(\gamma)m(\mu)$$

$$= m(\pi) \int \log(\frac{\mathrm{d}\gamma_1}{\mathrm{d}\mu}) \mathrm{d}\gamma_1 + m(\gamma)\mathrm{KL}(\pi_1|\mu) - m(\gamma)m(\mu)$$

$$= \int \left( \int \log(\frac{\mathrm{d}\gamma_1}{\mathrm{d}\mu}) \mathrm{d}\gamma_1 \right) \mathrm{d}\pi + m(\gamma)\mathrm{KL}(\pi_1|\mu) - m(\gamma)m(\mu),$$

and also from $\mathrm{KL}(\pi_1|\mu) = \int \log(\frac{\mathrm{d}\gamma_1}{\mathrm{d}\mu}) \mathrm{d}\gamma_1 - (m(\gamma) - m(\mu))$. Similar formulas hold for $(\pi_2, \gamma_2)$ and $(\pi, \gamma)$. Summing all KL terms yields the expression for $c_{\gamma}^{\varepsilon}$.  $\square$

### D.4  Discrete setting and formulas

In order to implement those algorithms, one consider discrete mm-spaces $X = (x_i)_{i=1}^{n}$ and $Y = (y_j)_{j=1}^{m}$, endowed with discrete measures $\mu = \sum_i \mu_i \delta_{x_i}$ and $\nu = \sum_j \nu_j \delta_{y_j}$, where $\mu_i, \nu_j \geq 0$. The distance matrices are $D_{i,i'}^{X} \triangleq d_X(x_i, x_{i'})$ and $D_{j,j'}^{X} \triangleq d_X(y_j, y_{j'})$. Transport plans are thus also discrete $\pi = \sum_{i,j} \pi_{i,j} \delta_{(x_i, y_j)}$.

**Algorithm 2 – UGW($\mathcal{X}$, $\mathcal{Y}$, $\rho$, $\varepsilon$) in discrete form**

**Input:** mm-spaces $\mathcal{X} = (D^X_{i,j}, (\mu_i)_i)$ and $\mathcal{Y} = (D^Y_{i,j}, (\nu_j)_j)$, relaxation $\rho$, regularization $\varepsilon$
**Output:** approximation $(\pi, \gamma)$ minimizing 6

1: Initialize matrix $\pi_{i,j} = \gamma_{i,j} = \mu_i \nu_j / \sqrt{(\sum_i \mu_i)(\sum_j \nu_j)}$, vector $g_j^{(s=0)} = 0$.
2: **while** $\pi$ has not converged **do**
3:     Update $\pi \leftarrow \gamma$
4:     Define $m(\pi) \leftarrow \sum_{i,j} \pi_{i,j}$, $\tilde{\rho} \leftarrow m(\pi)\rho$, $\tilde{\varepsilon} \leftarrow m(\pi)\varepsilon$
5:     Define $c \leftarrow \text{ComputeCost}(\mathcal{X}, \mathcal{Y}, \pi, \rho, \varepsilon)$
6:     **while** $(f, g)$ has not converged **do**
7:         $f \leftarrow -\frac{\tilde{\varepsilon}\tilde{\rho}}{\tilde{\varepsilon}+\tilde{\rho}} \log \left[ \sum_j \exp \left( (g_j - c_{i,j})/\tilde{\varepsilon} + \log \nu_j \right) \right]$
8:         $g \leftarrow -\frac{\tilde{\varepsilon}\tilde{\rho}}{\tilde{\varepsilon}+\tilde{\rho}} \log \left[ \sum_i \exp \left( (f_i - c_{i,j})/\tilde{\varepsilon} + \log \mu_i \right) \right]$
9:     **end while**
10:     Update $\gamma_{i,j} \leftarrow \exp \left[ (f_i + g_j - c_{i,j})/\tilde{\varepsilon} \right] \mu_i \nu_j$
11:     Rescale $\gamma \leftarrow \sqrt{m(\pi)/m(\gamma)}\gamma$
12: **end while**
13: Return $(\pi, \gamma)$.

The functional $\mathcal{L}$ now reads in this discrete setting

$$\int (d_X(x,x') - d_Y(y,y'))^2 \mathrm{d}\pi(x,y)\mathrm{d}\pi(x',y') = \sum_{i,j,k,\ell} (D^X_{i,j} - D^Y_{k,\ell})^2 \pi_{i,k}\pi_{j,\ell},$$

and $\quad \text{KL}(\pi_1 \otimes \pi_1 | \mu \otimes \mu) = \sum_{i,j} \log\left(\frac{\pi_{1,i}\pi_{1,j}}{\mu_i\mu_j}\right)\pi_{1,i}\pi_{1,j} - \sum_{i,j} \pi_{1,i}\pi_{1,j} + \sum_{i,j} \mu_i\mu_j$

$$= 2m(\pi)\sum_i \log\left(\frac{\pi_{1,i}}{\mu_i}\right)\pi_{1,i} - m(\pi)^2 + m(\mu)^2,$$

where we define the marginals $\pi_{1,k} \triangleq \sum_j \pi_{k,j}$, $\pi_{2,\ell} \triangleq \sum_i \pi_{i,\ell}$ and $m(\pi) = \sum_{i,j} \pi_{i,j}$.

When one runs the stabilized implementation of Sinkhorn's iterations with a ground cost $C_{i,j} = C(x_i, y_j)$ between the points, it is necessary to use a Log-Sum-Exp reduction which reads

$$f_i \leftarrow -\frac{\varepsilon\rho}{\varepsilon+\rho}\text{LSE}_j\left[(g_j - C_{i,j})/\varepsilon + \log(\mu_j)\right] \tag{31}$$

where $\text{LSE}_j$ is a reduction performed on the index $j$. It reads

$$\text{LSE}_j(C_{i,j}) \triangleq \log\left(\sum_j \exp(C_{i,j} - \max_k C_{i,k})\right) + \max_k C_{i,k}, \tag{32}$$

where the logarithm and exponential are pointwise operations.

We also provide an algorithm that computes the cost $c_\pi^\varepsilon$ defined in Proposition (10). We focus on the case $D_\varphi = \rho\text{KL}$ and $\lambda(t) = t^2$ which is computable with complexity $O(n^3)$ as shown in Peyré et al. [2016]. Indeed, note that one has

$$\int (d_X(x,x') - d_Y(y,y'))^2 \mathrm{d}\pi(x',y') = \int d_X(x,x')^2 \mathrm{d}\pi_1(x') + \int d_Y(y,y')^2 \mathrm{d}\pi_2(y')$$

$$- 2\int d_X(x,x')d_Y(y,y')\mathrm{d}\pi(x',y').$$

**Algorithm 3 – ComputeCost($\mathcal{X}, \mathcal{Y}, \pi, \rho, \varepsilon$) in discrete form**

---

**Input:** mm-spaces $\mathcal{X} = (D_{i,j}^X, (\mu_i)_i)$ and $\mathcal{Y} = (D_{k,\ell}^Y, (\nu_j)_j)$, transport matrix $(\pi_{j,k})_{j,k}$, relaxation $\rho$, regularization $\varepsilon$

**Output:** cost $c_\pi^\varepsilon$ defined in Proposition 10

1: Compute $\pi_{1,j} \leftarrow \sum_k \pi_{j,k}$ and $\pi_{2,k} \leftarrow \sum_j \pi_{j,k}$ $\{\pi_1 = \pi\mathbf{1}$ and $\pi_2 = \pi^\top\mathbf{1}\}$
2: Compute $A_i \leftarrow \sum_j (D_{i,j}^X)^2 \pi_{1,j}$ $\{A = (D^X)^{\circ 2}\pi_1\}$
3: Compute $B_\ell \leftarrow \sum_k (D_{k,\ell}^Y)^2 \pi_{2,k}$ $\{B = (D^Y)^{\circ 2}\pi_2\}$
4: Compute $C_{i,\ell} \leftarrow \sum_j D_{i,j}^X \left(\sum_k D_{k,\ell}^Y \pi_{j,k}\right)$ $\{C = D^X\pi D^Y\}$
5: Compute $E \leftarrow \rho \sum_j \log\left(\frac{\pi_{1,j}}{\mu_j}\right)\pi_{1,j} + \rho \sum_k \log\left(\frac{\pi_{2,k}}{\nu_k}\right)\pi_{2,k} + \varepsilon \sum_{j,k} \log\left(\frac{\pi_{jk}}{\mu_j\nu_k}\right)\pi_{j,k}$
6: Return $c_{\pi,i,\ell}^\varepsilon \leftarrow A_i + B_\ell - 2C_{i,\ell} + E$

---

# E    Supplementary experiments

We provide in this section details on Section 4. We start with supplementary synthetic experiments illustrating various features of UGW. We present our approach to approximate the distance CGW using a bi-convex relaxation and alternate minimization. We prove the tightness of this relaxation and provide details on the experiments of Section 3. Then we provide details on the PU learning experiments.

## E.1    Synthetic experiments

**Robustness to outlier**    Figure 4 shows another experiment on a 2-D dataset, using the same display convention as in Figure 1. It corresponds to the two moons dataset with additional outliers (displayed in cyan). Decreasing the value of $\rho$ (thus allowing for more mass creation/destruction in place of transportation) is able to reduce and even remove the influence of the outliers, as expected. Furthermore, using small values of $\rho$ tends to favor "local structures", which is a behavior quite different from UW (1). Indeed, for UW, $\rho \to 0$ sets to zero all the mass of $\pi$ outside of the diagonal (points are not transported), while for UGW, it is rather pairs of points with dissimilar pairwise distances which cannot be transported together.

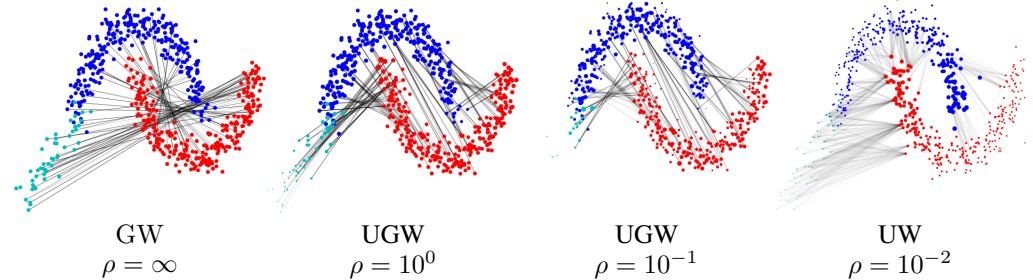

| GW | UGW | UGW | UW |
|---|---|---|---|
| $\rho = \infty$ | $\rho = 10^0$ | $\rho = 10^{-1}$ | $\rho = 10^{-2}$ |

Figure 4: GW and UGW applied to two moons with outliers. A matching using UW is provided to display how invariance to isometries is encoded in the matching.

**Graph matching and comparison with Partial-GW.**    We now consider two graphs $(X, Y)$ equipped with their respective geodesic distances. These graphs correspond to points embedded in $\mathbb{R}^2$, and the length of the edges corresponds to their Euclidean length. These two synthetic graphs are close to be isometric, but differ by addition or modification of small sub-structures. The colors $c(x)$ are defined on the "source" graph $X$ and are mapped by an optimal plan $\pi$ on $y \in Y$ to a color $\frac{1}{\pi_1(y)} \int_X c(x) \mathrm{d}\pi(x, y)$. This allows to visualize the matching induced by GW and UGW for a varying $\rho$, as displayed in Figure 5. The graphs for GW should be taken as reference since there is no mass creation. The POT library [Flamary and Courty, 2017] is used to compute GW.

For large values of $\rho$, UGW behaves similarly to GW, thus producing irregular matchings which do not preserve the overall geometry of the shapes. In sharp contrast, for smaller values of $\rho$ (e.g. $\rho = 10^{-1}$), some fine scale structures (such as the target's small circle) are discarded, and UGW is able to produce a meaningful partial matching of the graphs. For intermediate values ($\rho = 10^0$), we observe that the two branches and the blue cluster of the source are correctly matched to the target, while for GW the blue points are scattered because of the marginal constraint.

Figure 6 shows a comparison with Partial-GW [Chapel et al., 2020], computed using the POT library. It is close to UGW with a $\mathrm{TV}^\otimes$ penalty, since partial OT is equivalent to the use of a TV relaxation of the marginal. UGW with a $\mathrm{KL}^\otimes$ penalty is first computed for a given $\rho$, then the total mass $m$ of the optimal plan is computed, and is used as a parameter for PGW which imposes this total mass as a constraint. Figure 5 and 6 display the transportation strategy associated to both methods. KL-UGW operates smooth transitions between transportation and creation of mass, while PGW either performs pure transportation or pure destruction/creation of mass. In Figure 6 nodes of the graphs are removed and thus ignored by the matching. Note also that since PGW is equivalent to solving GW on sub-graphs, the color distribution of GW and PGW are similar.

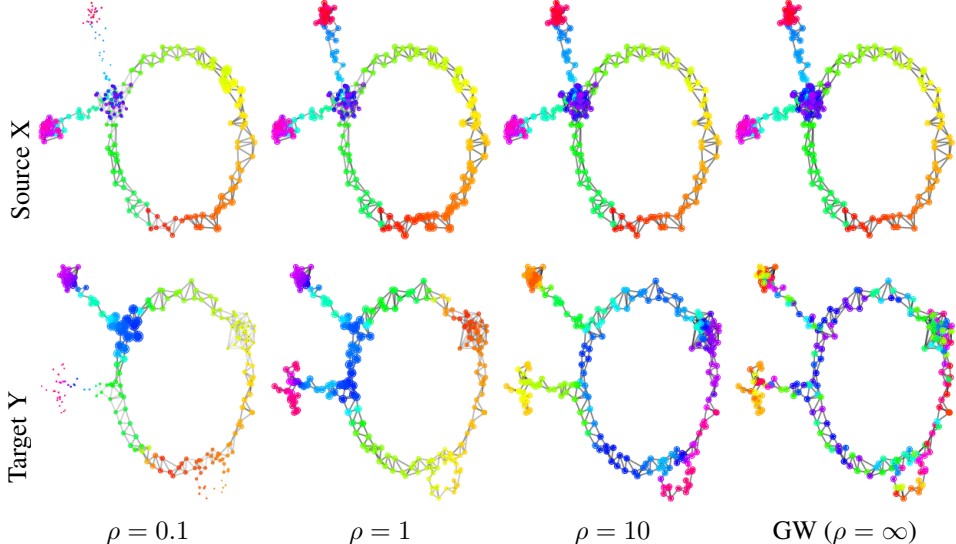

Figure 5: Comparison of UGW and GW for graph matching.

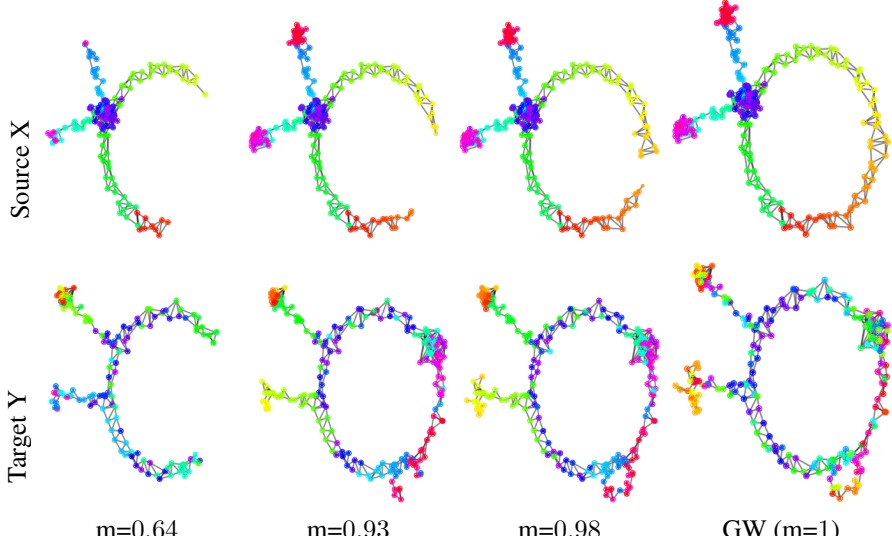

Figure 6: Comparison of Partial-GW for graph matching. Here $m$ is the budget of transported mass.

**Influence of $\varepsilon$.** Figures 1, 4, 5 and 6 do not show the influence of $\varepsilon$. This parameter is set of a low value $\varepsilon = 10^{-2}$ on a domain $[0,1]^2$ so as to approximate the optimal plan of the unregularized UGW problem. We present now an experiment on graphs which highlights the impact of $(\varepsilon, \rho)$ on the plan $\pi$.

We compare two graphs $(\mathcal{X}, \mathcal{Y})$ displayed Figure 7. The graph $\mathcal{X}$ is composed of two communities of equal size connected with random edges. The graph $\mathcal{Y}$ is similar to $\mathcal{X}$, but the communities are imbalanced and it contains outliers. Moving inside a community costs 1, reaching another community costs 4 and reaching an outlier 2. We equip the mm-space with uniform weights and shortest path distance.

We plot in Figure 8 optimal transport plans $\pi$ for given values of $(\varepsilon, \rho)$, including the balanced case $GW_\varepsilon$ where $\rho = \infty$. The transport matrix has a block structure: the 2 horizontal blocks correspond to $\mathcal{X}$ and its two communities, the 4 vertical blocks corresponds to $\mathcal{Y}$ (with, from left to right, the large blue community, the small red one, then the pink and green outliers). Decreasing $\rho$ results in a more structured transport matrix: outliers are removed and inter-community matching is avoided. Again,

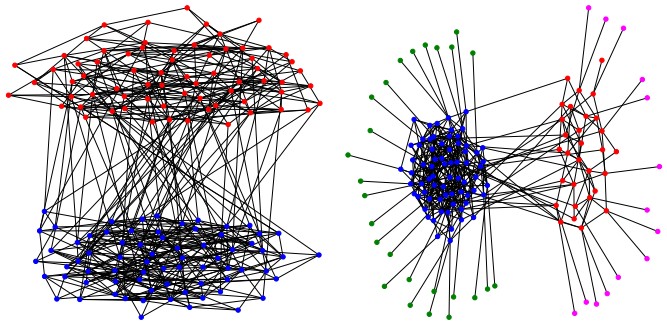

Figure 7: Graphs $\mathcal{X}$ (left) and $\mathcal{Y}$ (right) plotted using networkx.

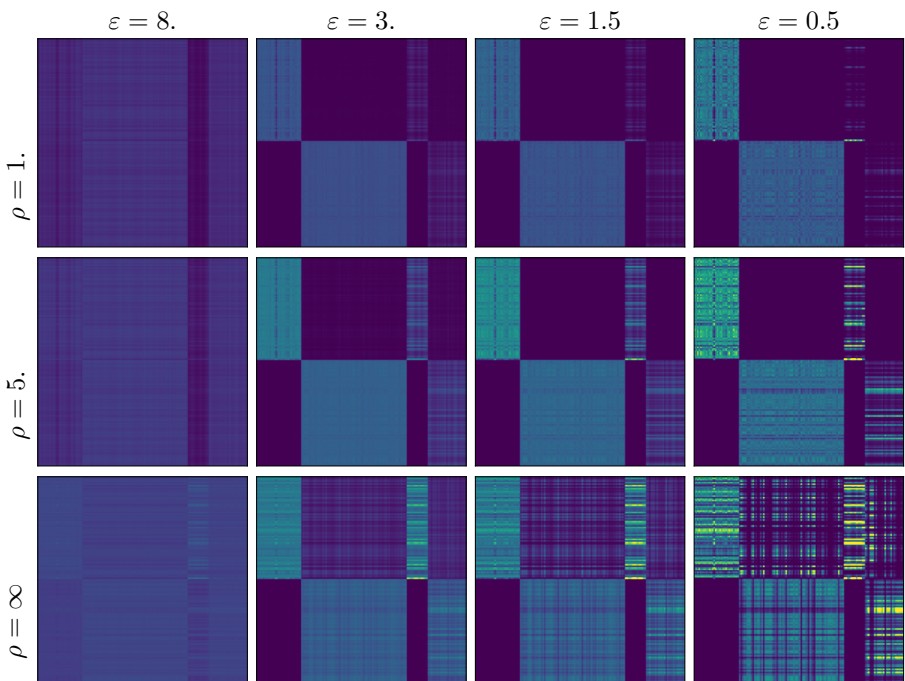

Figure 8: Display of the optimal transport plan $\pi$. The color scale is common to all plots.

the marginal constraint of $\mathrm{GW}_\varepsilon$ makes the plan more sensitive to structural noise (e.g. outliers) in graphs. Concerning the parameter $\varepsilon$, increasing it creates correlations between pairs of points whose distortion is of order of $\sqrt{\varepsilon}$. Indeed, we see for $\varepsilon = 3$ that correlation between communities and their outliers appear, even for small $\rho$. Furthermore, when $\varepsilon$ is too large the transport becomes uninformative, which highlights a crucial trade-off between computational speed and expressiveness of the transport plan.

## E.2 Computation of the CGW distance

In this section we focus on computing the distance CGW (4), which is a quadratic minimization program with linear constraint. Similar to what is performed with UGW 3, we consider a relaxation using a tensorized conic plan $\alpha \otimes \beta$ with $\alpha, \beta \in \mathcal{U}_p(\mu, \nu)$. The minimized cost thus reads

$$\mathcal{H}(\alpha, \beta) \triangleq \int \mathcal{D}([d_X(x, x'), rr'], [d_Y(y, y'), ss'])^q \, \mathrm{d}\alpha([x, r], [y, s]) \mathrm{d}\beta([x', r'], [y', s']). \quad (33)$$

Note that for fixed $\beta \in \mathcal{U}_p(\mu, \nu)$, the minimization w.r.t. $\alpha$ is a convex linear program with the linear conic constraint set $\mathcal{U}_p(\mu, \nu)$ and with cost

$$\mathcal{C}_{\mathfrak{C}}(x, r, y, s) \triangleq \int \mathcal{D}([d_X(x, x'), rr'], [d_Y(y, y'), ss'])^q \, d\beta([x', r'], [y', s']). \tag{34}$$

Since we focus on the numerical implementation of CGW, we consider the setting of Gaussian-Hellinger distance which computes the distortion with $\lambda(t) = t^2$, due to a reduced memory and computation complexity to calculate $|d_X - d_Y|^2$ (see Section 3). In that case the cone distance reads for a given $\rho$

$$\mathcal{D}([d_X(x, x'), rr'], [d_Y(y, y'), ss'])^2 = \rho\Big[(rr')^2 + (ss')^2 - 2rr'ss'\, e^{-|d_X - d_Y|^2/2\rho}\Big]. \tag{35}$$

Before focusing on the discretization of this problem to make it computable, we prove that when $|d_X - d_Y|^2$ is a conditionnaly definite kernel then the above cost is a negative kernel on $\mathcal{U}_p(\mu, \nu)$. Thus Theorem 2 holds.

**Proposition 11.** *Assume that the kernel $|d_X - d_Y|^2$ is conditionnaly negative definite. Then the cost* (35) *is a negative definite kernel on $\mathcal{U}_p(\mu, \nu)$.*

*Proof.* Take any plan $\alpha \in \mathcal{U}_p(\mu, \nu)$. Integrating against $(rr')^2$ or $(ss')^2$ yields a constant term. Indeed one has for $(rr')^2$

$$\int (rr')^2 d\alpha([x, r], [y, s]) d\alpha([x', r'], [y', s']) = \left(\int (r)^2 d\alpha([x, r], [y, s])\right)^2$$

$$= \left(\int d\mu(x)\right)^2$$

$$= m(\mu)^2.$$

Thus minimizing w.r.t. (35) is equivalent to minimizing w.r.t. $-2rr'ss'\, e^{-|d_X - d_Y|^2/2\rho}$, which is a product of positive definite kernels $(rr')$, $(ss')$ and $e^{-|d_X - d_Y|^2/2\rho}$ thanks to Berg's Theorem Berg et al. [1984] and because we assume the kernel $|d_X - d_Y|^2$ is c.n.d. Due to the extra minus sign we get that the kernel is negative definite, which ends the proof. $\qquad\square$

An important point is to implement the constraint set $\mathcal{U}_p(\mu, \nu)$ which integrates against radial coordinates $(r, s) \in \mathbb{R}_+^2$. Such integration is impossible in practice, but thanks to Liero et al. [2015, Theorem 7.20], we know that the radius can be restricted to $[0, R]$ where $R^2 = m(\mu)^2 + m(\nu)^2$ (up to a dilation of the plan). Thus we propose to discretize the constraint by sampling regularly the interval as $\{lR/L, l \in [\![0, L]\!]\}$.

We consider discrete mm-spaces as in Section D, i.e. mm-spaces noted as $\mathcal{X} = (D_{i,j}^X, (\mu_i)_i)$ and $\mathcal{Y} = (D_{i,j}^Y, (\nu_j)_j)$. Write a conic plan $\alpha_{ijkl} = \alpha([x_i, r_k], [y_j, s_l])$. The conic constraints read for $k \in [\![0, K]\!]$ and $l \in [\![0, L]\!]$

$$\sum_{j,k,l} \Big(\frac{kR}{K}\Big)^2 \alpha_{ijkl} = \mu_i \quad \text{and} \quad \sum_{i,k,l} \Big(\frac{lR}{L}\Big)^2 \alpha_{ijkl} = \nu_j.$$

The cost $\mathcal{C}_{\mathfrak{C}}$ (34) is computed via the formula

$$\mathcal{C}_{ijkl} \triangleq \sum_{i',j',k',l'} \rho\Bigg[\Big(\frac{kR}{K}\frac{k'R}{K}\Big)^2 + \Big(\frac{lR}{L}\frac{l'R}{L}\Big)^2 - 2\Big(\frac{kR}{K}\frac{k'R}{K}\frac{lR}{L}\frac{l'R}{L}\Big)e^{-|D_{i,i'}^X - D_{j,j'}^Y|^2/2\rho}\Bigg]\alpha_{i'j'k'l'}.$$

Eventually, the whole program solving one step of the alternate minimization algorithm is given Equation (36). The approximation of CGW is performed by alternatively updating $\alpha$ and $\mathcal{C}_{\mathfrak{C}}$ until the minimization attains a local minima

$$\min_{\alpha_{ijkl}} \left\{\sum_{i,j,k,l} \mathcal{C}_{ijkl}\alpha_{ijkl} \quad \text{s.t.} \quad \sum_{j,k,l}\Big(\frac{kR}{K}\Big)^2\alpha_{ijkl} = \mu_i \quad \text{and} \quad \sum_{i,k,l}\Big(\frac{lR}{L}\Big)^2\alpha_{ijkl} = \nu_j\right\}. \tag{36}$$

| Dataset | # of samples | # of positives | Dim. | PCA Dim. |
|---------|-------------|----------------|------|----------|
| *-caltech | 1,123 | 151 | surf: 800 / decaf: 4096 | surf: 10 / decaf: 40 |
| *-amazon | 958 | 92 | surf: 800 / decaf: 4096 | surf: 10 / decaf: 40 |
| *-webcam | 295 | 29 | surf: 800 / decaf: 4096 | surf: 10 / decaf: 40 |
| *-dslr | 157 | 12 | surf: 800 / decaf: 4096 | surf: 10 / decaf: 40 |

Table 2: Characteristics of datasets

**Details on the experiments of Section 4.** One can observe in the above procedure that the memory complexity of $\alpha$ and $\mathcal{C}_{\mathfrak{C}}$ is prohibitively high to use it in practice, due to the discretization of the radial coordinate which make the size of both tensors scaling as $O(NMKL)$ where $N, M$ are the number of samples in the spaces $(\mathcal{X}, \mathcal{Y})$. Thus our experiment are performed considering Euclidean mm-spaces composed of samples $N, M \in \{2, 3, 5\}$, and we take $K = L = 10$. To guarantee as much as possible that we reach the global minima, we consider 10 random initializations and 10 random permutation matrices $P$ lifted as conic plan by setting $\alpha_{..kl} = P$ for any $(k, l)$. The latter initialization is assumed to be close to extremal points of the constraint polytope. Since Theorem 2 holds for Euclidean mm-spaces, the optimal plan is also an extremal point of the polytope. To compare CGW with UGW, we set a solver with a level of entropy $\varepsilon = 10^{-3}$. In Figure 3 we set $\rho = 10^{-1}$.

### E.3 Details on PU learning experiments

**Details on training for PU learning tasks.** We present the characteristics of the datasets in Table 2. The variance of the accuracy results presented in Table 1 is presented in Table 4. The computations were made on an internal GPU cluster composed of 10 Tesla K80 and 3 Tesla P100. We also detail the parameters of the numerical solver computing UGW which is the core component of our numerical experiments.

- The maximum number of iteration to update the plan is set to 3000.
- The tolerance on convergence of $\pi$ in log-scale is set to $10^{-5}$, i.e. the algorithm stops when $\left\|\log \pi^{t+1} - \log \pi^t\right\|_\infty < tol$.
- The maximum number of iteration to update the Sinkhorn potentials is set to 3000.
- The tolerance on convergence of $(f, g)$ is set to $10^{-6}$, i.e. the algorithm stops when $\left\|f^{t+1} - f^t\right\|_\infty < tol$.

**Initialization for cross-domain tasks.** To initialize UGW when the features are different we propose to use a UOT solution of a matching between distance histograms which reads

$$\text{FLB}(\mathcal{X}, \mathcal{Y}) \triangleq \min \int_{X \times Y} |\bar{\mu} \star d_X - \bar{\nu} \star d_Y|^2 \mathrm{d}\pi + \rho \text{KL}(\pi_1|\mu) + \rho \text{KL}(\pi_2|\nu) + \varepsilon \text{KL}(\pi|\mu \otimes \nu), \quad (37)$$

where $\mu \star d_X(x) \triangleq \int d_X(x, x') \mathrm{d}\mu(x')$ is the eccentricity, i.e. a histogram of aggregated distances, and $\bar{\mu} = \mu / m(\mu)$. In Mémoli [2011] this relaxation is refered as FLB and is a lower bound of GW, but in our unbalanced setting this program cannot a priori be compared with UGW.

**Reducing the number of parameters.** In Table 1, the accuracy for UGW is performed by selecting a pair of parameters $(\rho_1, \rho_2)$ for each task via a validation protocol detailed Section 4. It is desirable to reduce the number of parameters, to see if the performance does not significantly decrease, and avoid overparameterization of the task. We propose in this section two strategies The first case keeps one pair $(\rho_1, \rho_2)$ over all tasks. The second case keeps a pair for each pair of domain tasks (i.e. surf↔surf, decaf↔decaf, surf↔decaf and decaf↔surf) for a total of 8 parameters, which allows to normalize adaptively each dataset via an adapted choice of parameters $(\rho_1, \rho_2)$. The validation protocol is modified since we aggregate accuracies from different tasks. The selected parameters are obtained by taking the highest mean excess accuracy over all tasks, where the excess is defined by comparing the accuracy to the case where we only predict false positives. This measure of performance is computed on the validation folds, and we report the accuracy over the testing folds in Table 3.

| Dataset | prior | Init (PW) | PGW | **UGW** | UGW (2 param.) | UGW (8 param.) |
|---|---|---|---|---|---|---|
| surf-C → surf-C | 0.1 | **89.9** | 84.9 | 83.9 | 81.8 | 83.9 |
| surf-C → surf-A | 0.1 | 81.8 | 82.2 | **83.5** | 83.1 | 83.3 |
| surf-C → surf-W | 0.1 | **81.9** | 81.3 | 80.3 | 80.1 | 80.4 |
| surf-C → surf-D | 0.1 | 80.0 | 81.4 | **83.2** | 80.3 | **83.2** |
| surf-C → surf-C | 0.2 | **79.7** | 75.7 | 75.4 | 67.5 | 75.4 |
| surf-C → surf-A | 0.2 | 65.6 | 66.0 | **76.4** | 74.0 | 73.0 |
| surf-C → surf-W | 0.2 | 65.1 | 64.3 | **67.3** | 63.8 | 64.9 |
| decaf-C → decaf-C | 0.1 | **93.9** | 83.0 | 86.8 | 84.8 | 84.8 |
| decaf-C → decaf-A | 0.1 | 80.1 | 81.4 | **85.6** | 83.7 | 83.7 |
| decaf-C → decaf-W | 0.1 | 80.1 | 82.7 | **86.1** | 85.6 | 85.6 |
| decaf-C → decaf-D | 0.1 | 80.6 | **83.8** | 83.4 | 83.6 | 83.6 |
| decaf-C → decaf-C | 0.2 | **90.6** | 76.7 | 80.5 | 75.7 | 75.7 |
| decaf-C → decaf-A | 0.2 | 62.5 | 68.7 | 74.7 | **75.0** | **75.0** |
| decaf-C → decaf-W | 0.2 | 65.7 | 75.9 | 79.2 | **80.2** | **80.2** |
| Dataset | prior | Init (FLB) | PGW | **UGW** | UGW (2 param.) | UGW (8 param.) |
| surf-C → decaf-C | 0.1 | 85.0 | 85.1 | **85.6** | 85.0 | 85.0 |
| surf-C → decaf-A | 0.1 | 84.2 | **87.1** | 83.6 | 83.5 | 83.5 |
| surf-C → decaf-W | 0.1 | 86.2 | **88.6** | 86.8 | 87.4 | 87.4 |
| surf-C → decaf-D | 0.1 | 84.7 | **91.1** | 90.7 | 89.3 | 89.3 |
| surf-C → decaf-C | 0.2 | 74.8 | 75.6 | 75.9 | **76.2** | **76.2** |
| surf-C → decaf-A | 0.2 | 76.2 | **87.9** | 82.4 | 83.2 | 83.2 |
| surf-C → decaf-W | 0.2 | 81.5 | 88.4 | **89.9** | 88.8 | 88.8 |
| decaf-C → surf-C | 0.1 | 81.7 | 81.0 | 81.1 | 81.9 | **82.1** |
| decaf-C → surf-A | 0.1 | 80.9 | 81.2 | **82.4** | 81.2 | 82.1 |
| decaf-C → surf-W | 0.1 | 82.0 | 81.3 | **83.5** | 80.8 | 80.7 |
| decaf-C → surf-D | 0.1 | 80.0 | 80.8 | **81.5** | 80.0 | **81.5** |
| decaf-C → surf-C | 0.2 | 66.6 | 63.7 | 65.2 | 66.5 | **67.9** |
| decaf-C → surf-A | 0.2 | 62.9 | 62.4 | **69.3** | 62.2 | 68.5 |
| decaf-C → surf-W | 0.2 | 65.1 | 61.4 | **83.3** | 61.1 | 65.0 |

Table 3: Accuracy for all tasks. The left block are domain adaptation experiments with similar features, where both PGW and UGW are initialised with PW. The right block are domain adaptation experiments with different features, and the reported init is FLB (see Appendix E) used for UGW.

| Dataset | prior | Init (PW) | PGW | **UGW** | UGW (2 param.) | UGW (8 param.) |
|---|---|---|---|---|---|---|
| surf-C → surf-C | 0.1 | 2.05 | 1.95 | 2.93 | 2.14 | 2.94 |
| surf-C → surf-A | 0.1 | 1.25 | 1.89 | 2.14 | 2.29 | 3.33 |
| surf-C → surf-W | 0.1 | 1.33 | 1.82 | 0.73 | 0.45 | 0.82 |
| surf-C → surf-D | 0.1 | 0.00 | 1.69 | 2.63 | 0.73 | 2.63 |
| surf-C → surf-C | 0.2 | 2.98 | 4.66 | 5.07 | 2.42 | 5.07 |
| surf-C → surf-A | 0.2 | 2.87 | 3.29 | 3.59 | 2.15 | 9.46 |
| surf-C → surf-W | 0.2 | 1.95 | 2.12 | 9.22 | 1.82 | 7.61 |
| decaf-C → decaf-C | 0.1 | 1.61 | 2.24 | 2.46 | 1.64 | 1.64 |
| decaf-C → decaf-A | 0.1 | 0.44 | 1.91 | 4.52 | 2.08 | 2.08 |
| decaf-C → decaf-W | 0.1 | 0.44 | 2.55 | 1.65 | 1.90 | 1.90 |
| decaf-C → decaf-D | 0.1 | 0.92 | 1.54 | 2.06 | 1.67 | 1.67 |
| decaf-C → decaf-C | 0.2 | 2.54 | 3.59 | 5.73 | 2.53 | 2.54 |
| decaf-C → decaf-A | 0.2 | 2.09 | 4.39 | 7.46 | 4.52 | 4.52 |
| decaf-C → decaf-W | 0.2 | 1.93 | 3.60 | 5.89 | 3.61 | 3.61 |
| Dataset | prior | Init (PW) | PGW | **UGW** | UGW (2 param.) | UGW (8 param.) |
| surf-C → decaf-C | 0.1 | 2.79 | 2.64 | 3.01 | 2.71 | 2.71 |
| surf-C → decaf-A | 0.1 | 2.08 | 6.50 | 3.28 | 2.82 | 2.82 |
| surf-C → decaf-W | 0.1 | 1.89 | 5.63 | 3.97 | 3.62 | 3.62 |
| surf-C → decaf-D | 0.1 | 1.93 | 8.09 | 7.09 | 7.46 | 7.46 |
| surf-C → decaf-C | 0.2 | 2.56 | 3.32 | 4.02 | 3.66 | 3.66 |
| surf-C → decaf-A | 0.2 | 3.74 | 6.61 | 10.5 | 8.04 | 8.04 |
| surf-C → decaf-W | 0.2 | 2.75 | 5.82 | 3.33 | 3.64 | 3.64 |
| decaf-C → surf-C | 0.1 | 1.82 | 1.61 | 1.21 | 1.77 | 2.29 |
| decaf-C → surf-A | 0.1 | 1.18 | 1.94 | 2.11 | 1.36 | 2.10 |
| decaf-C → surf-W | 0.1 | 1.67 | 2.03 | 3.94 | 1.01 | 1.17 |
| decaf-C → surf-D | 0.1 | 0.00 | 1.60 | 1.70 | 0.00 | 1.70 |
| decaf-C → surf-C | 0.2 | 3.04 | 2.92 | 7.21 | 3.24 | 4.08 |
| decaf-C → surf-A | 0.2 | 1.84 | 4.54 | 5.92 | 2.04 | 5.19 |
| decaf-C → surf-W | 0.2 | 2.86 | 3.23 | 6.43 | 1.52 | 3.76 |

Table 4: Standard deviation of accuracy for all tasks of Figure 1. The left block are domain adaptation experiments with similar features, where both PGW and UGW are initialised with PW. The right block are domain adaptation experiments with different features, and the reported init is FLB (37) used for UGW.