# OpenReview forum: "The Unbalanced Gromov Wasserstein Distance: Conic Formulation and Relaxation"
_NeurIPS.cc/2021/Conference — NeurIPS 2021 Poster_

### Official Review · Reviewer_XuXr · 2021-07-15

**Rating:** 7
**Confidence:** 4

**Summary:**

The paper extends the Gromov-Wasserstein (GW) distance framework to two novel relaxations, the unbalanced GW (UGW) and the conic GW (CGW), which both address the question of comparing arbitrary positive measures on different metric spaces. The paper also proves that CGW is upper bounded by UGW, and introduces an algorithm to compute UGW based on Sinkhorn iterations. Numerical experiments are presented to illustrate the performance of the new algorithm.

**Ethical Concerns:**

None.

**Limitations And Societal Impact:**

None.

**Main Review:**

The paper presents new results of theoretical and practical importance for the two proposed GW formulations. The text is well-written and streamline. Technical writing is also to a high standard, with clean notation and coherent proofs.

The results are quite strong, combining ideas from unbalanced optimal transport (UOT), conic optimal transport (COT) and GW. By choosing a specific divergence in the new formulations, the paper partly retrieves the well-known equality connecting UOT and COT. The tightness result based on maximization of convex functional furthers the theoretical understanding of GW and elucidates how to extend the Sinkhorn algorithm to the GW setting.

Some specific comments/concerns I have are as follows:
1. After Theorem 2 it is claimed that the kernel $|d_X-d_Y|^2$ is negative semi-definite. According to the proof of Theorem 2 it is supposed to be negative semi-definite on the set of all signed measures due to the unbalanced setting. (Vayer, Titouan, et al. "Sliced Gromov-Wasserstein”) show that it is negative semi-definite on the set of all couplings, thus accounting for the balanced GW distance. Do the authors have a reference/proof for this general claim? This is crucial for the usefulness of Theorem 2 and to applications of the proposed methods.

2. The paper would benefit from more account of theoretical aspects of CGW. The reviewer understands that this may exceed the scope of the current submission, but more results on, e.g., induced topology on mm-spaces and its relation to GW, would be a nice addition. The authors should at least consider mentioning such research trajectories as future directions.

3. The entropic regularization is standard, but in UGW setting it remains unclear whether all the nice properties (convergence, global optimality, stopping criterion, etc.) still hold. For completeness, these should be checked, in addition to just showing the computational cost of each step. The dependence of the entropic relaxation on $\epsilon$ should also be examined, e.g., if it converges to the UGW as $\epsilon\to 0$. In general the 'doubly entropic' GW is worth studying even for the balanced case.

4. The paper repeatedly claims that the regular GW is recovered as $\rho\to\infty$, which should also be verified rigorously as it is unclear whether it’s the distance or the optimal coupling that’s recovered, and 'recovered' in what sense.

5. The main theorem claims UGW $\geq$ CGW, and the paper only illustrates numerically how tight this inequality can be. Is it possible to bound the gap and to give criterions when they are equal?

**Time Spent Reviewing:**

5

---

> ### Author Response · Authors · 2021-08-10
> **Response to Reviewer XuXr**
>
> We thank the reviewer for its remarks. Below are quotations of his review followed by our answers.
>
> > After Theorem 2 it is claimed that the kernel is negative semi-definite. According to the proof of Theorem 2 it is supposed to be negative semi-definite on the set of all signed measures due to the unbalanced setting. [Vayer et al.] show that it is negative semi-definite on the set of all couplings, thus accounting for the balanced GW distance. Do the authors have a reference/proof for this general claim? This is crucial for the usefulness of Theorem 2 and to applications of the proposed methods.
>
> The reviewer is correct, and this was a mistake in our statement, we thank the reviewer for pointing this out. Indeed, it holds that the distortion kernel is conditionally negative on the set of plans with marginals equal to zero when both metrics are also conditionally negative kernels. Thus to have our result valid for UGW, we need to prove that the optimal plans $(\pi,\gamma)$ of the biconvex relaxation have the same marginals, such that the kernel is negative for $\pi - \gamma$.
>
> At the moment we did not find a proof of this point, but we are working on it. Without a proof that our result holds for UGW, we must clarify its applicability in the revised version. However, our generalization of Konno’s result remains true (see Theorem 3, Appendix D), which is in any case a contribution on the properties of QAPs. Furthermore, Theorem 2 holds for $GW_\epsilon$ (since the marginals are fixed), which is a new extension to the result of Konno (which applies only for $GW$ without entropy).
>
>
> > The paper would benefit from more account of theoretical aspects of CGW. The reviewer understands that this may exceed the scope of the current submission, but more results on, e.g., induced topology on mm-spaces and its relation to GW, would be a nice addition. The authors should at least consider mentioning such research trajectories as future directions.
>
> The questions on extra properties of CGW (completeness, topology, geodesics) or UGW (is it also a distance ?) are indeed left as open questions for future works. We will mention those on the paper.
>
> > The entropic regularization is standard, but in UGW setting it remains unclear whether all the nice properties (convergence, global optimality, stopping criterion, etc.) still hold. For completeness, these should be checked, in addition to just showing the computational cost of each step.
>
> We acknowledge that very little is known about the convergence of the alternate minimization scheme. Using results from [Tseng 2001], it can be shown that all cluster points are stationary. One can also relate this algorithm to the classical DC (difference of convex) method and to mirror descent (as shown in [Solomon et al 2016]), which might provide some intuition about why in practice it always converges.
>
> > The dependence of the entropic relaxation on should also be examined, e.g., if it converges to the UGW as $\epsilon\rightarrow 0$.
>
>
> > The paper repeatedly claims that the regular GW is recovered as $\rho\rightarrow\infty$, which should also be verified rigorously as it is unclear whether it’s the distance or the optimal coupling that’s recovered, and 'recovered' in what sense.
>
> The answer to the asymptotics $\epsilon\rightarrow 0$ and $\rho\rightarrow 0$ are similar. We will add details on this, and provide here a sketch of proof for $\epsilon\rightarrow 0$.
>
> Take two fixed spaces $(X,Y)$ and a sequence $\epsilon_n$ going to zero, and a sequence of optimal plans $\pi_n$ for $UGW_{\epsilon_n}(X,Y) = L_{\epsilon_n}(\pi_n)$. Under the setting of Proposition 1, one can assume the sequence lies in a compact space (with Banach-Alaoglu theorem). We can obtain first convergence of the functional value: the lsc property of the functional yields a lower bound on the limit when $\epsilon_n\rightarrow 0$, and the optimal plan $\pi^*$ for $UGW(X,Y)=L(\pi^*)$ is suboptimal for $UGW_{\epsilon_n}(X,Y)$, thus we obtain $L_{\epsilon_n}(\pi_n)\leq L_{\epsilon_n}(\pi^*)$ and $L_{\epsilon_n}(\pi^*) \rightarrow UGW(X,Y)$, hence the equality of values. Concerning the convergence of minimizers, we have in the compact setting that subsequences converge to global minimizers, due to the non-convexity of UGW. If we assume uniqueness of the minimizer for UGW (e.g. when there are no symmetries), then all subsequences converge to the same limit, and the whole sequence converges to the global minimizer.
>
> > The main theorem claims $UGW \geq CGW$, and the paper only illustrates numerically how tight this inequality can be. Is it possible to bound the gap and to give criterions when they are equal?
>
> This is indeed a very important question. Figure 2 suggests that for small shift (i.e. close mm-spaces) both costs are close. Thus a first step, which is a reasonnable goal for future work, is to show that they define the same “geodesic space”.

---

> > ### Comment · Reviewer_XuXr · 2021-08-19
> > **Answer to author rebuttal**
> >
> > First, I would like to thank authors for their answers to my questions.
> >
> > I view the paper quite favorably and appreciate the promised revisions. However, it currently seems that the result of Theorem 2 is unsalvageable, which weakens the contribution of the paper. I therefore to change my score to 7.

---

### Official Review · Reviewer_abgG · 2021-07-17

**Rating:** 6
**Confidence:** 3

**Summary:**

This paper proposes two unbalanced Gromov-Wasserstein formulations, namely Unbalanced Gromov-Wasserstein (UGW) divergence and Conic Gromov-Wasserstein (CGW) distance. Both of them are positive and definite. UGW has a scalable and GPU-friendly algorithm, which is applicable to large learning problems. CGW is a distance between mm-spaces up to isometry. It is theoretically and empirically shown that UGW can be used as a surrogate upper-bounding CGW. The authors then apply UGW to positive unlabeled learning with Caltech office dataset and achieve comparable or better performance than Chapel et al., 2020.

**Limitations And Societal Impact:**

This paper considers the formulation of unbalanced Gromov-Wasserstein distance, and does not contain any specific applications that may have negative societal impact.

**Main Review:**

This paper flows well. That being said, I would strongly recommend the authors to simplify the notations and terminologies in the main paper, especially in the introduction section. This can make the work more accessible for general readers. Besides, notations like $\iota$ and $\bot$ lack proper definition.

It is an interesting idea to use a scalable divergence to surrogate a real distance.

Although according to the algorithm the computation of UHW should be efficient and scalable, it would be better if the computation time is also reported.

I am curious that whether the plots for $\rho=1, 10$ will have the same trend as $\rho=0.1$ is the shift is larger.

Line 228 the code at https://github.com/anonymous-conference-submission does not exist (by July 16. This is just a minor point because there is code in the supplement material).


**Time Spent Reviewing:**

2.5

---

> ### Author Response · Authors · 2021-08-10
> **Response to Reviewer abgG**
>
> We thank you for your remarks. We are surprised by the low grade which was given, and hope you will reconsider your evaluation. We hope to have addressed all the issues you raised, and would be happy to discuss further if needed to improve the clarity. The other reviewers are very positive and we hope you will be as well after this rebuttal !
>
> You will find below quotations of your reviews followed by our answers.
>
> > I would strongly recommend the authors to simplify the notations and terminologies in the main paper, especially in the introduction section.
>
> We will make efforts to improve the pedagogy of the introduction, and give some more details and examples when introducing notation
>
> > Besides, notations like $\iota$ and lack $\bot$ proper definition.
>
> Indeed, we will add the definitions of these notations. $\iota_C$ is the indicator of a set $C$ ($0$ on $C$ and $+\infty$ outside). $\mu^\bot$ is the orthogonal part of the Lebesgue decomposition with respect to some measure $\nu$. Intuitively it corresponds to the part of $\mu$ which is defined outside of the support of $\nu$.
>
> > I am curious that whether the plots for $\rho = 1, 10$ will have the same trend as $\rho = 0.1$  if the shift is larger.
>
> Indeed, there is a similar trend for larger values of $\rho$. What matters is the magnitude of the distortion, which increases with the shift $t$. Thus the gap will appear for a larger shift when $\rho$ is larger. We will update the plot to show this behavior.
>
> > Line 228 the code at (...) does not exist.
>
> It is a placeholder link to guarantee the anonymity of the submission. The link of the repository would be revealed after the reviewing process.

---

### Official Review · Reviewer_mUoT · 2021-07-18

**Rating:** 7
**Confidence:** 3

**Summary:**

The paper deals with unbalanced Gromov-Wasserstein (UGW)  to compare un-registered metric measure spaces not necessarily endowed with a probability distribution. It proposes the relaxation of the  usual marginal constraints using quadratic  divergences leading to  a new  divergence that can be used to compare metric measure spaces.  By  adding an adapted entropic regularization to  the UGW formulation, an alternate optimization scheme based on Sinkhorn iterations  is designed. The  paper also  introduces another conic  formulation for Unbalanced GW (CGW) that  defines a distance invariant to isometries. The former is shown  to upper-bound CGW which optimization procedure is computationally  heavy. Both CGW and UGW  are illustrated on simulated. Furthermore application of  UGW to PU  learning shows  the effectiveness  of the  proposed UGW divergence.

**Ethical Concerns:**

The  paper does not raise  any ethical issues.

**Limitations And Societal Impact:**

Unless I was mistaken, the paper does not discuss the broader impact of the proposed approaches. On my view the work does not raise specific issues except the computation burdent inheritent to  GW.

**Main Review:**

- Overall, the paper is well written and well  organized. The rationale behind the proposed method is justified and the  novelty brought  by  the approach is stated clearly. Theoretical  analysis of the proposed unbalanced  GW formulations is  provided  along  with the strengths and  limitations of UGW and CGW. Optimization approach to compute both GW-based divergences is also given. Empirical evaluations support the  effectiveness of the  proposed methods. Hence,  the paper brings a significant and interesting contribution to the field of unbalanced Gromov-Wasserstein distance.
- The main novelty  of the paper is  the definition of Unbalanced GW based $\varphi$-quadratic divergences that allow to express a 2-homogeneous relaxation of  the  marginal constraints of balanced GW. The paper  should deserve a few lines to provide better intuition on the quadratic divergences.
- GW  discrepancy is  known to   isometry-invariant. Is it the  case for UGW? As a general comment, the theoretical properties (beyond existence and definiteness) of UGW discrepancy measure should be more discussed and  made clearer.
-  Lemma 1  plays  a central role in connecting UGW to the conic formulation of GW. Such as presented, that Lemma is hard to handle with several dense  notations. The paper should improve the introduction and the comments of that Lemma for a better readability.
- Interestingly the conic formulation  (CGW) defines a distance up  to isometries, a desirable property for some practical applications.  Also a strong result  of the paper is that the CGW distance is upper-bounded by the UGW discrepancy measure. The latter being computationally more favorable, it offers a theoretically justified  way to approximate CGW. The paper investigates empirically  the tightness of the relation CGW $\leq$ UGW. Nevertheless the empirical analysis has to be extended to isometry-invariance framework.
- The  weak points  of the paper are related to the optimization  procedure. Indeed,  the more general setting of UGW is narrowed to some specific divergences (KL in fact) or map function $\lambda$  limiting the applicability scope of UGW. I guess such assumptions are unavoidable to allow competitive optimization schemes. The second remark is the use  of entropic regularization in order to elaborate an efficient  bi-convex relaxation  to solve UGW. Depending on the choice of the regularization parameter, the upper-bound on CGW  may be less tight as expected. Can the authors discuss that  ? Also, is there any result on the convergence of regularized UGW towards UGW when $\epsilon  \rightarrow  0$?
- The paper informs that CGW has a high computation burden. Can  the paper provide  some insights on the computational complexity or memory requirement to compute CGW?
- The empirical evaluations show the good behavior and  the benefits of the proposed unbalanced GW divergences. Especially the obtained results are impressive for some experimental setups (see Table 1) and  they illustrate the benefit of the proposed marginal constraint relaxation using quadratic-divergences. Compare to PGW  how  one can explain the effect of applying such quadratic-divergences in-lieu of  the  classical ones?
- In the robustness to imbalanced classes, it  might be interesting to  plot the matching suggested by PGW.

After feedback
-------------------
I thank the authors for providing detailed responses to the raised points. I highly appreciate their endeavour to clarify most concerns, especially the  isometry-invariance property of UGW, the computational burden of UGW in the general setting or the convergence convergence of regularized UGW towards UGW when $\epsilon  \rightarrow  0$.  I expect the revision will improve on these points and on the dense notations. Hence, I still have a positive view of the paper and I keep my rating.


**Time Spent Reviewing:**

9

---

> ### Author Response · Authors · 2021-08-10
> **Response to Reviewer mUoT**
>
> We thank the reviewer for its remarks. Below are quotations of his review followed by our answers.
>
> > The paper should deserve a few lines to provide better intuition on the quadratic divergences.
>
> From a “practical” perspective, some intuitions are given lines 147 to 152, 2-homogeneity is important to obtain a loss which is invariant (up to a scalar factor) when rescaling the spaces. From a theoretical perspective, this is also important to be able to prove the relation between UGW and CGW, and we will add some more details.
>
> > GW discrepancy is known to isometry-invariant. Is it the case for UGW?
>
> Indeed, similarly to GW, UGW is isometry-invariant in the sense that if $X \sim X’$ and $Y \sim Y’$, then $UGW(X, Y) = UGW(X’, Y’)$. Thus UGW compares equivalence classes of mm-spaces (for the isometry relation $\sim$). The proof is obtained by considering optimal plans $\pi$ and $\pi’$ for $(X,Y)$ and $(X’,Y’)$. Since the pushforward of $\pi$ via the isometries is suboptimal for $(X’,Y’)$, it yields $UGW(X, Y) \geq UGW(X’, Y’)$. The converse inequality holds with $\pi’$, hence the equality. We will add this property in the submission.
>
> The proof is similar for CGW, thus CGW also compares equivalence classes for the isometry relation $\sim$. We will explain this in the revised version.
>
> > the theoretical properties (beyond existence and definiteness) of UGW discrepancy measure should be more discussed and made clearer
>
> We will refactor the main properties of UGW (such as the one mentioned in the previous point) in a proposition instead of being expressed as a remark.
>
> > Such as presented, that Lemma 1 is hard to handle with several dense notations.
>
> We agree on this point. This Lemma is fundamental for the connection between UGW and CGW, and deserves more space to expand the notations. We will provide more details on this in the final version.
>
> > Nevertheless the empirical analysis has to be extended to isometry-invariance framework.
>
> We would like to raise the attention of the reviewer on the fact that UGW and CGW are both isometry-invariant by construction (see the point above), and as a consequence, all numerical computations of both divergences share this invariance. The only “missing” property for UGW is the triangular inequality. We will further insist on this in the revised version.
>
> > The weak points of the paper are related to the optimization procedure. Indeed, the more general setting of UGW is narrowed to some specific divergences (KL in fact) or map function $\lambda$ limiting the applicability scope of UGW.
>
> We acknowledge this is a weak point. It would still be possible to solve UGW for other divergences, but the numerical computation would be much more costly (in terms of both time and memory, see lines 265-267 in the paper). Computing for other divergences (such as TV) is left for future research.
>
>
> > Depending on the choice of the regularization parameter, the upper-bound on CGW may be less tight as expected.
>
> Indeed one has $CGW \leq UGW \leq UGW_\epsilon$. It is an open problem to theoretically assess the deviation between $UGW_\epsilon$ and $UGW$, but one could expect it to be similar to the one obtained for classical (regularized) optimal transport, which is of the order $O(\epsilon*\log(\epsilon))$, similar to the work of Christian Leonard. We will mention this. In the experiments of Figure 3 we took $\epsilon = 10^{-3}$ on the unit square (see Appendix E.2), which should guarantee that $UGW_\epsilon$ approximates well $UGW$.
>
> > is there any result on the convergence of regularized UGW towards UGW when $\epsilon\rightarrow 0$?
>
>  Indeed, we did not consider that question in the paper, and we will add a formal proof. We provide a sketch of proof below.
>
>  Take two fixed spaces $(X,Y)$ and a sequence $\epsilon_n$ going to zero, and a sequence of optimal plans $\pi_n$ for $UGW_{\epsilon_n}(X,Y) = L_{\epsilon_n}(\pi_n)$. Under the setting of Proposition 1, one can assume the sequence lies in a compact space (with Banach-Alaoglu theorem). We can obtain first convergence of the functional value: the lsc property of the functional yields a lower bound on the limit when $\epsilon_n\rightarrow 0$, and the optimal plan $\pi^*$ for $UGW(X,Y)=L(\pi^*)$ is suboptimal for $UGW_{\epsilon_n}(X,Y)$, thus we obtain $L_{\epsilon_n}(\pi_n)\leq L_{\epsilon_n}(\pi^*)$ and $L_{\epsilon_n}(\pi^*) \rightarrow UGW(X,Y)$, hence the equality of values. Concerning the convergence of minimizers, we have in the compact setting that subsequences converge to global minimizers, due to the non-convexity of UGW. If we assume uniqueness of the minimizer for UGW (e.g. when there are no symmetries), then all subsequences converge to the same limit, and the whole sequence converges to the global minimizer.
>
> > Can the paper provide some insights on the computational complexity or memory requirement to compute CGW?
>
> Some details on the computations are presented in Appendix E.2. The key limitations are the discretization of the radial coordinates (noted $(r,s)$ in Section 2.2) and the fact that the computation of the conic distance involves a 8D-tensor. For instance, with $N$ samples for both spaces and $K$ values to discretize the radial axis, the tensor has memory complexity $Q=O(N^4.K^4)$. If one uses an interior point solver, as we did for the numerical simulations, the  complexity is expected to be of the order $O(Q^3)$. Of course, it might be possible to develop much more efficient approximate solver, but we leave this for future woks.
>
> > Compare to PGW how one can explain the effect of applying such quadratic-divergences in-lieu of the classical ones?
>
> Indeed, PGW is not using a penalized total variation (TV) divergence but rather a total mass constraint. If one replaces this by a “classical” (non quadratic) TV divergence, a first issue detailed in lines 147-152 is that the resulting loss suffers from a lack of homogeneity (which we think is not a good feature, so that the resulting transport plan is invariant to total mass rescaling). The second issue is that one cannot anymore show the relation between UGW and CGW. Also, there does not seem to be any simple numerical scheme to solve for a penalized Total Variation divergence.
>
> > In the robustness to imbalanced classes, it might be interesting to plot the matching suggested by PGW
>
> We will add this experiment in the revised version.

---

> ### Comment · Reviewer_mUoT · 2021-08-23
> **After authors feedback**
>
> I would like to thank authors for their responses to the the raised points. Now my questions are addressed. I expect the revision will improve on the notations and clarify the various interesting properties of the proposed methods. Hence, I keep my original rating.

---

### Author Response · Authors · 2021-08-10
**Common response to the reviewers**

We wish to thank the reviewers for their relevant remarks which will improve the submission. We provide to each reviewer a dedicated answer below their review.

---

### Decision · Program_Chairs · 2021-09-27

**Decision:**

Accept (Poster)

**Comment:**

A standard approach for defining distances between metric measure spaces is to use the Gromov-Wasserstein distance. However this formalism is limited to settings where the metric measure space has an underlying probability distribution. The main contributions of this paper are in formulating unbalanced analogues of the Gromov-Wasserstein distance that work with general positive measures. The specific formulations adapt ideas from unbalanced and conic optimal transport. Furthermore they use entropic regularization to design a Sinkhorn-style iterative algorithm and give experimental results on synthetic data and for problems in domain adaptation. The paper is well-written and has a nice mix of contributions. The only weakness is that they specialize to KL divergences which seems to limit the scope and applicability. Furthermore, in their setup working with more general divergences appears to be much more computationally challenging.